# Witnessing the onset of reionization through Lyman-α emission at redshift 13

Joris Witstok[1,2,3,4 ✉], Peter Jakobsen[3,4], Roberto Maiolino[1,2,5], Jakob M. Helton[6], Benjamin D. Johnson[7], Brant E. Robertson[8], Sandro Tacchella[1,2], Alex J. Cameron[9], Renske Smit[10], Andrew J. Bunker[9], Aayush Saxena[5,9], Fengwu Sun[6,7], Stacey Alberts[6], Santiago Arribas[11], William M. Baker[1,2], Rachana Bhatawdekar[12], Kristan Boyett[9], Phillip A. Cargile[7], Stefano Carniani[13], Stéphane Charlot[14], Jacopo Chevallard[9], Mirko Curti[15], Emma Curtis-Lake[16], Francesco D'Eugenio[1,2,17], Daniel J. Eisenstein[7], Kevin N. Hainline[6], Gareth C. Jones[1,2,9], Nimisha Kumari[18], Michael V. Maseda[19], Pablo G. Pérez-González[11], Pierluigi Rinaldi[6], Jan Scholtz[1,2], Hannah Übler[1,2,20], Christina C. Williams[21], Christopher N. A. Willmer[6], Chris Willott[22] & Yongda Zhu[6]

Cosmic reionization began when ultraviolet (UV) radiation produced in the first galaxies began illuminating the cold, neutral gas that filled the primordial Universe[1,2]. Recent James Webb Space Telescope (JWST) observations have shown that surprisingly UV-bright galaxies were in place beyond redshift $z = 14$, when the Universe was less than 300 Myr old[3–5]. Smooth turnovers of their UV continua have been interpreted as damping-wing absorption of Lyman-α (Ly-α), the principal hydrogen transition[6–9]. However, spectral signatures encoding crucial properties of these sources, such as their emergent radiation field, largely remain elusive. Here we report spectroscopy from the JWST Advanced Deep Extragalactic Survey (JADES[10]) of a galaxy at redshift $z = 13.0$ that reveals a singular, bright emission line unambiguously identified as Ly-α, as well as a smooth turnover. We observe an equivalent width of $EW_{Ly-\alpha} > 40$ Å (rest frame), previously only seen at $z < 9$ where the intervening intergalactic medium becomes increasingly ionized[11]. Together with an extremely blue UV continuum, the unexpected Ly-α emission indicates that the galaxy is a prolific producer and leaker of ionizing photons. This suggests that massive, hot stars or an active galactic nucleus have created an early reionized region to prevent complete extinction of Ly-α, thus shedding new light on the nature of the earliest galaxies and the onset of reionization only 330 Myr after the Big Bang.

Using the Near-Infrared Camera (NIRCam[12]) and Mid-Infrared Instrument (MIRI[13]) aboard the JWST, we obtained deep imaging as part of the JADES and JADES Origins Field (JOF)[14] programmes. A careful search for high-redshift galaxy candidates exploiting the 14-band NIRCam coverage[15,16] led to the identification of JADES-GS+53.06475-27.89024 (JADES-GS-z13-1-LA hereafter) as the most robust redshift $z \gtrsim 11.5$ photometric candidate in the JOF based on its blue colour and clear 'dropout' signature, confidently rejecting a brown dwarf solution. Because the discontinuity strength (>20× in flux between the NIRCam F150W and F200W filters) further rules out a Balmer break resulting from an evolved stellar population at much lower redshift, the photometry strongly favours a solution at $z \approx 13$, at which Ly-α, the $2p \rightarrow 1s$ electronic transition of hydrogen, is shifted to 1.7 μm in the observed frame and any photons emitted at shorter wavelengths are completely absorbed by neutral hydrogen (H I) in the intervening intergalactic medium (IGM).

Follow-up spectroscopy of JADES-GS-z13-1-LA was obtained as part of JADES with the JWST Near-Infrared Spectrograph (NIRSpec)[17], principally in PRISM mode (exposure time of 18.7 h), covering wavelengths from 0.6 μm up to 5.3 μm at low resolution ($R \approx 100$). As shown in Fig. 1, the resulting spectrum unequivocally confirms the redshift to be $z \approx 13.0$ (Methods), even if the break is smooth rather than sharp, which indeed is expected for sources embedded in a highly neutral IGM owing to Ly-α damping-wing absorption[18], as has been seen directly in quasar

[1]Kavli Institute for Cosmology, University of Cambridge, Cambridge, UK. [2]Cavendish Laboratory, University of Cambridge, Cambridge, UK. [3]Cosmic Dawn Center (DAWN), Copenhagen, Denmark. [4]Niels Bohr Institute, University of Copenhagen, Copenhagen, Denmark. [5]Department of Physics and Astronomy, University College London, London, UK. [6]Steward Observatory, University of Arizona, Tucson, AZ, USA. [7]Center for Astrophysics | Harvard & Smithsonian, Cambridge, MA, USA. [8]Department of Astronomy and Astrophysics, University of California, Santa Cruz, Santa Cruz, CA, USA. [9]Department of Physics, University of Oxford, Oxford, UK. [10]Astrophysics Research Institute, Liverpool John Moores University, Liverpool, UK. [11]Centro de Astrobiología (CAB), CSIC-INTA, Madrid, Spain. [12]European Space Astronomy Centre (ESAC), European Space Agency (ESA), Madrid, Spain. [13]Scuola Normale Superiore, Pisa, Italy. [14]Institut d'Astrophysique de Paris, Sorbonne Université, CNRS, UMR 7095, Paris, France. [15]European Southern Observatory, Garching, Germany. [16]Centre for Astrophysics Research (CAR), Department of Physics, Astronomy and Mathematics, University of Hertfordshire, Hatfield, UK. [17]INAF – Osservatorio Astronomico di Brera, Milan, Italy. [18]AURA for European Space Agency, Space Telescope Science Institute (STScI), Baltimore, MD, USA. [19]Department of Astronomy, University of Wisconsin–Madison, Madison, WI, USA. [20]Max-Planck-Institut für extraterrestrische Physik, Garching, Germany. [21]NSF's National Optical-Infrared Astronomy Research Laboratory (NOIRLab), Tucson, AZ, USA. [22]NRC Herzberg, Victoria, British Columbia, Canada. ✉e-mail: joris.witstok@nbi.ku.dk

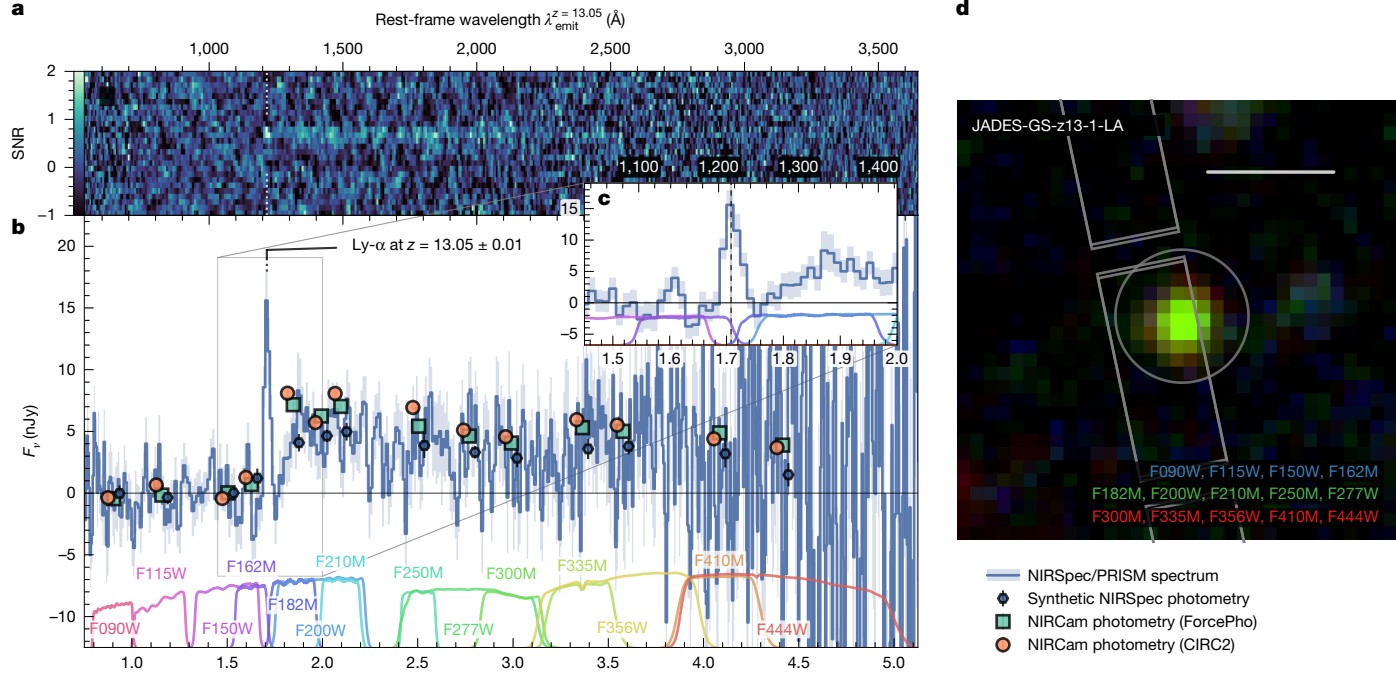

**Fig. 1 | NIRCam and NIRSpec/PRISM observations of JADES-GS-z13-1-LA.**
**a**, Two-dimensional SNR map of the PRISM spectrum (not used for extraction of the one-dimensional spectrum; see Methods for details). **b**, One-dimensional sigma-clipped PRISM spectrum (uncorrected for further path losses; see Methods) and photometric measurements (slightly offset in wavelength for visualization) according to the legend at the bottom right. Synthetic photometry is obtained by convolving the spectrum with the filter transmission curves shown at the bottom. Shading and error bars represent $1\sigma$ uncertainty. **c**, Zoom-in on the emission line at 1.7 μm, which falls precisely between the F162M and F182M medium-band filters. **d**, False-colour image of JADES-GS-z13-1-LA constructed by stacking NIRCam filters for each colour channel as annotated. The placement of the NIRSpec microshutters, nearly identical across the two visits, is shown in grey, as is the circular 0.3″-diameter extraction aperture for the CIRC2 photometry. A physical scale of 1 kpc (0.28″ at $z = 13.05$) is indicated as the scale bar.

spectra[19]. Spectra of $z \gtrsim 9$ galaxies recently discovered by the JWST have also hinted at the existence of IGM damping wings[3,7], although many cases have been observed to far exceed pure IGM absorption, which has been ascribed to local damped Ly-α (DLA) absorbing systems (column densities $N_{HI} > 10^{20.3}$ cm$^{-2}$; ref. 20) interpreted as pockets of dense, neutral gas within or near the galaxy[6,8,9,21].

Notably, unlike any other $z > 10$ galaxies confirmed by the JWST[3–6,22–24], the PRISM spectrum also reveals a bright emission line detected at high signal-to-noise ratio (SNR = 6.4) and consistently across the two independent visits (Methods). Located at the blue edge of the spectral break, it is observed at $\lambda_{obs} = 1.7084 \pm 0.0014$ μm and, although the continuum directly underneath is not detected, we can conservatively place a lower limit on the rest-frame EW of >40 Å. The only viable explanation, considering the clear break and the absence of nearby foreground sources and any other lines (Methods), is to identify the line as Ly-α at a redshift of $z_{Ly\text{-}\alpha} = 13.05 \pm 0.01$. However, owing to the resonant nature of Ly-α, we note that the systemic redshift is probably slightly lower.

If not arising from collisional excitation, expected to be subdominant even at interstellar medium (ISM) densities of $n \approx 10^4$ cm$^{-3}$ (ref. 25), this immediately implies that JADES-GS-z13-1-LA produces a substantial number of ionizing Lyman continuum (LyC) photons as quantified by the production efficiency, for which we find a robust lower limit of $\xi_{ion} \gtrsim 10^{25.1}$ Hz erg$^{-1}$ (Methods). Although already close to the canonical value required for star-forming galaxies to complete reionization[26], this value increases considerably if any Ly-α photons are absorbed within the galaxy or scattered out of our line of sight in the IGM. This should be a notable effect at $z = 13$, as the Universe is still highly neutral[7,19], even if a local ionized 'bubble' around the galaxy facilitates the transmission of Ly-α photons[11]. Note that, although photon diffusion by means of resonant scattering off neutral gas in the IGM is predicted

to result in extended Ly-α halos around galaxies before reionization[27], such diffuse emission cannot explain the observed line properties. From non-detections in our medium-resolution spectra, although less sensitive than the PRISM, we do however infer that the line is probably broadened spectrally (Methods).

Fitting a variety of standard stellar population synthesis (SPS) models to the observed spectral energy distribution (SED) of JADES-GS-z13-1-LA yields a young (10–20 Myr) and metal-poor (<2% Solar) stellar population, with little to no dust obscuration (Supplementary information). However, commonly used SED fitting codes do not have the capability to model the peculiar coexistence of Ly-α emission together with a smooth spectral turnover. To better understand its origin in JADES-GS-z13-1-LA, we therefore performed detailed spectral modelling in which we take into account potential absorption by DLA absorbers, transmission through a neutral, mean-density IGM with a local ionized bubble and instrumental effects such as path losses and the line spread function. For our fiducial model, we opt for a power-law continuum that offers the flexibility to recreate the steep UV slope, which from the NIRCam and NIRSpec data we consistently measure to be $\beta_{UV} \lesssim -2.7$ (Methods). However, we also considered the inclusion of nebular continuum, as the two-photon (2γ) continuum in lower-redshift galaxies has been suggested[28,29] as the potential origin of a UV turnover and Ly-α emission qualitatively similar to JADES-GS-z13-1-LA. Best-fitting models with a pure power-law and 2γ continuum are shown in Fig. 2.

Regardless of the choice of continuum, our model indicates that, across a range of reasonable emergent Ly-α profiles, approximately 5–10% of flux may be transmitted through the IGM, implying an intrinsic Ly-α luminosity of $L_{Ly\text{-}\alpha} \approx 3 \times 10^{43}$ erg s$^{-1}$. Here we allow for a non-zero LyC escape fraction causing a local ionized bubble with radius $R_{ion} \approx 0.2$ physical Mpc (pMpc) to form within an otherwise neutral IGM,

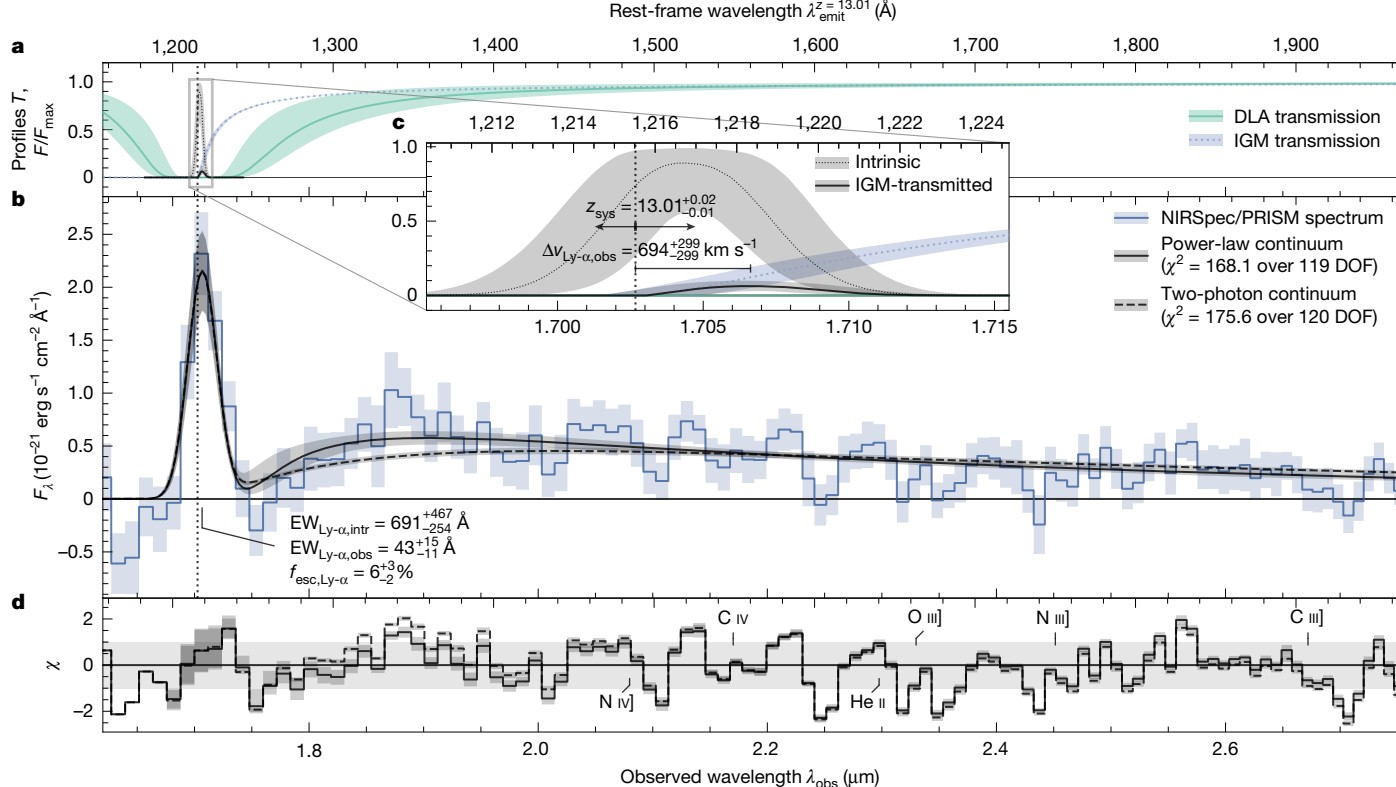

**Fig. 2 | Model of NIRSpec/PRISM observations of JADES-GS-z13-1-LA.**
**a**, Model curves for the IGM and DLA transmission $T$ (according to the legend on the right) and normalized Ly-α line profiles (see panel **c**). **b**, Blue line shows the sigma-clipped PRISM spectrum corrected for path losses (Methods). Model spectra with a power-law continuum, attenuated by DLA absorption, and a pure 2γ continuum are shown by the solid and dashed black lines, respectively. The legend shows their $\chi^2$ goodness-of-fit statistics compared with the degrees of freedom (DOF; Methods). The intrinsic and observed Ly-α EWs (relative to an unattenuated power-law continuum) and their ratio (the escape fraction) are annotated. **c**, Zoom-in on the intrinsic (dotted black line) and IGM-transmitted (solid black line) Ly-α line profiles. The vertical black dotted line shows the median systemic Ly-α redshift in the default model (Methods), differing from the Ly-α redshift by the observed velocity offset $\Delta v_{\text{Ly-α,obs}}$. **d**, For the two different models, $\chi$ represents the residuals normalized by the observational uncertainty of a single wavelength bin (diagonal elements of the covariance matrix). The location of other rest-frame UV lines are indicated, although none are significantly detected (Methods). Shading represents 1σ uncertainty on all lines.

without which the required luminosity would triple, a scenario disfavoured by the non-detection in the MIRI/F770W filter containing Hβ (Methods). Still, we find that the models consistently require $\xi_{\text{ion}} \approx 10^{26.5}$ Hz erg$^{-1}$, to either create the transmission-enhancing bubble or boost the intrinsic luminosity. For any appreciable IGM transmission, the observed Ly-α peak should fall substantially redwards ($\Delta v_{\text{Ly-α,obs}} \gtrsim$ 500 km s$^{-1}$) of the systemic redshift[22,30], which we therefore infer to be $z_{\text{sys}} = 13.01^{+0.02}_{-0.01}$.

For standard stellar models, the remarkably high $\xi_{\text{ion}}$ is untenable[31,32] under common initial mass functions (IMFs). Because $\xi_{\text{ion}}$ is directly sensitive to the hottest stars, its extreme value may be ascribed to an extension of the IMF to very massive stars[33,34]. The high average ionizing-photon energy of a $T = 10^5$ K blackbody moreover yields a two times higher ratio of Ly-α to LyC photons than standard case B recombination[25], thereby bringing the true $\xi_{\text{ion}}$ more closely in agreement with the theoretical stellar maximum[35]. One particularly intriguing class of objects predicted to radiate up to 40% of their bolometric luminosity as Ly-α are entirely metal-free Population III (Pop III) stars[36–38] thought to reach substantially higher masses and effective temperatures than subsequent metal-enriched stellar populations. However, the absolute UV magnitude of JADES-GS-z13-1-LA, $M_{\text{UV}} \approx -18.7$ mag, would require a stellar mass of $M_* \approx 10^6 M_\odot$ as a pure Pop III system, slightly higher than typical predictions[39]. Furthermore, the absence of strong He II λ 1,640 Å emission (Methods) may argue against the Pop III scenario[40], although its strength rapidly evolves several million years after a star-formation burst[37,38].

The presence of extraordinarily hot stars ($T_{\text{eff}} > 10^5$ K) required to explain such high $\xi_{\text{ion}}$ could naturally lead relatively low-density gas ($n \lesssim 10^4$ cm$^{-3}$) to emit a prominent nebular continuum with a UV turnover[29,41]. However, we find that, compared with the pure 2γ continuum, which only becomes further reddened by free-bound continuum emission at higher densities, the current data are better reproduced by a steep power law (Fig. 2). A scenario in which Ly-α emission is produced together with 2γ continuum as cooling radiation by means of collisional excitation in the dense core of a collapsing cloud[42] is therefore also disfavoured. The extremely blue UV continuum ($\beta_{\text{UV}} \lesssim -2.7$) consistently leads our models to prefer near-unity LyC escape fraction to reproduce the blue SED of JADES-GS-z13-1-LA, even with an IMF extending to 300 $M_\odot$ (Methods). Moreover, recent stellar models show[43] that the effective temperatures of very massive stars stagnate beyond 100 $M_\odot$, suggesting that a high LyC escape fraction remains necessary. Although this would corroborate the suggestion that JADES-GS-z13-1-LA is located inside an ionized bubble and could suppress He II, it still leaves the UV turnover to be explained.

Instead, the spectrum of JADES-GS-z13-1-LA therefore seems to necessitate notable DLA absorption ($N_{\text{HI}} \approx 10^{22.8}$ cm$^{-2}$ for the power-law continuum), as seen in several $z > 10$ galaxies[6,8,9,21]. If the DLA absorber were co-located with the galaxy, a specific geometry is required to simultaneously accommodate the escape of Ly-α and, potentially, LyC. As illustrated in Fig. 3, an inhomogeneous ISM or an edge-on disk and associated ionization cone may cause DLA absorption in compact continuum sources, which is circumvented by Ly-α emission[44].

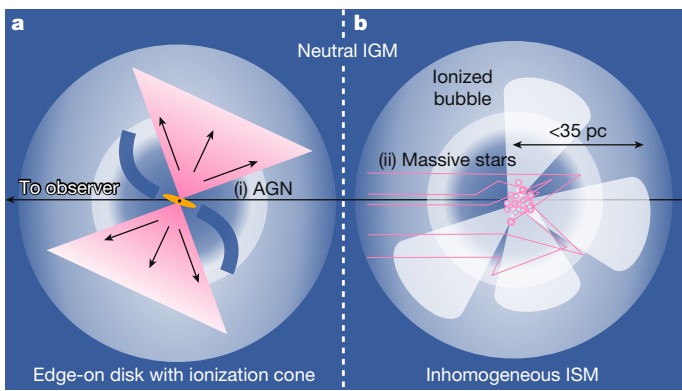

**Fig. 3 | Schematic of production, escape and absorption of Ly-α in JADES-GS-z13-1-LA. a,b,** Ly-α emission is indicated in pink, whereas dark blue shows H I gas. We identify two potential explanations each for the source of emission ((i) and (ii)) and modes of Ly-α modulation. **a,** An extended disk of neutral gas seen in edge-on orientation may cause DLA absorption of the continuum source, whereas an ionization cone perpendicular to the disk plane allows Ly-α photons to escape. Under this escape mechanism, the source of the Ly-α emission may be interchanged from an AGN (i) to a nuclear starburst (ii). **b,** Alternatively, if neutral gas in the ISM is inhomogeneously distributed, resonant scattering could allow Ly-α to diffuse outwards while the central source remains obscured by H I gas, as seen in local, compact, star-forming galaxies (see text for details).

Especially in the absence of dust, Ly-α emission could escape through resonant scattering while also becoming broadened in velocity space, consistent with observations. Empirically, Ly-α emission superimposed on DLA absorption has not only been reported for nearby UV-bright star-forming galaxies in which it has been interpreted as a sign of ISM inhomogeneity[45] but also in the case of active galactic nuclei[46].

Indeed, an accreting supermassive black hole may offer a comprehensive alternative explanation for the observed properties of JADES-GS-z13-1-LA. Effectively unresolved by NIRCam, its half-light radius of $\lesssim$35 pc (Methods) is smaller than most $z > 10$ galaxies[3,5,22,23]. Active galactic nuclei have been observed[47] to reach UV slopes much steeper than the standard thin-disk model[48] with $\beta_{UV} = -7/3 \approx -2.33$, as expected for a truncated accretion disk. They are also found[49] to have high LyC escape fractions and the broad Ly-α line could be linked to active galactic nucleus-driven outflows or a broad line region. Constraints on the at present undetected He II and other UV lines (Methods) are consistent with model predictions for metal-poor active galactic nuclei[40], altogether making JADES-GS-z13-1-LA a viable candidate.

Whether the Ly-α emission of JADES-GS-z13-1-LA originates in stars or a supermassive black hole, it reveals the rather extreme character of one of the earliest galaxies known, despite having been found in a modest survey area[16] examining a comoving volume of 50,000 Mpc³ between $z = 11$ and $z = 15$. At only 330 Myr after the Big Bang, the probable presence of a reionized region around this relatively UV-faint source readily constrains the timeline of cosmic reionization, favouring an early and gradual process driven (initially) by low-mass galaxies[50]. Furthermore, it provides tangible evidence for the Wouthuysen–Field coupling of the spin temperature of neutral hydrogen to that of the gas by means of the emission of Ly-α photons, the global evolution of which is anticipated to be uncovered soon by H I 21-cm experiments[51] to provide a complementary view of cosmic dawn.

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

## Methods

### Cosmology and conventions

A flat $\Lambda$CDM cosmology is used throughout based on the latest results of the Planck Collaboration[52], with $H_0 = 67.4$ km s$^{-1}$ Mpc$^{-1}$, $\Omega_m = 0.315$ and $\Omega_b = 0.0492$. The cosmic hydrogen fraction is fixed to $f_H = 0.76$. At $z = 13$, the Hubble flow is $H(z = 13) \approx 1,990$ km s$^{-1}$ Mpc$^{-1}$ and on-sky separations of $1''$ and $1'$ correspond to 3.53 physical kpc and 0.212 pMpc, respectively. We quote magnitudes in the AB system[53], emission-line wavelengths in vacuum and EWs in the rest frame unless explicitly mentioned otherwise.

### NIRCam observations and target selection

In the following sections, we describe the main JWST and auxiliary Hubble Space Telescope (HST) observations underlying this work. We refer to refs. 16,54 for details on the NIRCam and MIRI imaging, respectively, whereas ref. 5 provides a detailed description of the NIRSpec spectroscopy. Further details on the JADES survey strategy and data reduction are discussed in the survey overview paper[10] and the data release papers[55–57].

The NIRCam[12], MIRI[13] and NIRSpec[17,58] measurements presented in this work are associated with JWST Guaranteed Time Observations (GTO) programme IDs (PIDs) 1180 (PI: Eisenstein), 1210, 1286 and 1287 (PI: Luetzgendorf), further complemented with the JOF programme[14] (PID 3215; PIs: Eisenstein and Maiolino). Also, because the JOF itself is located within the Great Observatories Origins Deep Survey South (GOODS-S[59]) extragalactic legacy field, HST Legacy Field imaging[60] is publicly available, covering 0.4 μm to 1.8 μm between the Advanced Camera for Surveys (ACS) and Wide Field Camera 3 (WFC3).

Further MIRI imaging in the F770W filter was obtained[54] as coordinated parallel observations to JADES NIRCam observations (PID 1180). Several high-redshift targets, selected by refs. 15,16 based on the NIRCam images in the JOF, including JADES-GS-z13-1-LA (located at right ascension of +53.06475° and declination of −27.89024°), were followed up using the NIRSpec Micro-Shutter Assembly (MSA[61]) as part of PID 1287, scheduled between 10 and 12 January 2024.

### NIRSpec observations and data reduction

The NIRSpec observations spanned three consecutive visits. However, during visit 2, the lock on the guide star was lost, preventing it from being carried out nominally. Although different MSA configurations were used across visits, JADES-GS-z13-1-LA was observed in both visits 1 and 3 in the PRISM/CLEAR grating-filter combination (simply 'PRISM' hereafter) with resolving power of $30 \lesssim R \lesssim 300$ between wavelengths of 0.6 μm and 5.3 μm, as well as in the medium-resolution grating-filter combinations G140M/F070LP, G235M/F170LP and G395M/F290LP ('R1000 gratings'), each with resolving power $R \approx 1,000$. A sequence of exposures following three nod positions was repeated four times for each visit in PRISM mode and once for each of the R1000 gratings. Each nod sequence had an exposure time of 8,403.2 s, consisting of six integrations made up of 19 groups in NRSIRS2 readout mode[62]. Altogether, JADES-GS-z13-1-LA was observed for 67,225.6 s by the NIRSpec/PRISM and 16,806.4 s in each of the R1000 gratings.

We used version 3.1 of the data-reduction pipeline developed by the ESA NIRSpec Science Operations Team[61] and the NIRSpec GTO team (simply 'pipeline' hereafter), which produces flux-calibrated spectra largely following the algorithms used in the Space Telescope Science Institute (STScI) pipeline. We refer to previous works[3,5,56,57] for detailed descriptions of the NIRSpec data-reduction pipeline, an overview of which is given in ref. 61. In brief, three adjacent microshutters were opened to obtain background-subtracted spectra of individual sources, for which the subtraction follows a three-point nodding scheme discussed above. Initial path-loss corrections were calculated under the assumption of a point-source light profile placed at the same intra-shutter location of the source. The PRISM spectra take up an irregular wavelength grid with sampling such that the wavelength-dependent line spread function[17] always spans a fixed number of wavelength bins. Our fiducial ('sigma-clipped') spectrum combines all available sub-exposures in the three nodding positions, for which one-dimensional spectra are extracted over the central three spatial pixels (corresponding to 0.3″), through a custom sigma-clipping algorithm (see Supplementary information for details).

### Photometric measurements

We obtained photometric measurements of JADES-GS-z13-1-LA using two methods. Our fiducial photometry is determined using ForcePho (B.D.J. et al., manuscript in preparation) on all 14 available NIRCam filters (see also ref. 16), whereas MIRI/F770W follows a customized procedure following J.M.H. et al. (manuscript in preparation), both discussed in more detail below. An alternative approach to ForcePho is to measure fluxes in circular apertures with a diameter of 0.3″ ('CIRC2'). These results are summarized in Extended Data Table 1. We include CIRC2 photometry in the available HST bands, which, together with NIRCam filters up to and including F150W, are statistically fully consistent with non-detections ($\chi^2 = 11.6$ over ten filters, that is, $P = 0.31$).

Given that the full width at half maximum (FWHM) of the MIRI/F770W point spread function (PSF) is much larger than those of NIRCam[54], we considered the F444W–F770W colour of JADES-GS-z13-1-LA after convolving the F444W mosaic with the F770W PSF and rebinning to the F770W pixel size. We measured this colour assuming a circular aperture with 0.7″ diameter ('CIRC5'), which roughly corresponds to the 65% encircled energy of F770W, before applying aperture corrections. The reported MIRI/F770W flux is then inferred from the difference between the total CIRC5 NIRCam/F444W flux and the F444W–F770W colour. Using this approach, we are taking advantage of the higher spatial resolution afforded by NIRCam compared with MIRI. However, this measurement does not yield a significant detection ($F_\nu = 1.60 \pm 2.23$ nJy). Neglecting contributions from the [O III] $\lambda$ 4,960, 5,008 Å lines and underlying continuum, the MIRI non-detection would be consistent with an H$\beta$ flux of $F_{H\beta} \lesssim 6.7 \times 10^{-19}$ erg s$^{-1}$ cm$^{-2}$ ($3\sigma$), translating to an intrinsic Ly-$\alpha$ flux of $F_{Ly\text{-}\alpha} \lesssim 1.6 \times 10^{-17}$ erg s$^{-1}$ cm$^{-2}$ (case B recombination; for example, ref. 11).

To explore the morphology of JADES-GS-z13-1-LA, we first fitted Sérsic[63] profiles separately to the various available NIRCam filters (using the mosaic images) using the pysersic code[64]. We do not find a strong wavelength dependency of the morphology. In the F277W filter, which explores rest-frame wavelengths around $\lambda_{emit} \approx 2,000$ Å at $z = 13$, we constrain JADES-GS-z13-1-LA to have a half-light radius of $17.5^{+3.0}_{-1.7}$ mas and a Sérsic[63] index consistent with $n = 1$. This size approaches half the pixel size (that is, 15 mas) and should hence be treated as an upper limit, given that the mosaicing procedure probably introduces artificial smoothing.

To fit to independent dithered NIRCam exposures, we performed further modelling with ForcePho (B.D.J. et al., manuscript in preparation), assuming a model with a single intrinsic Sérsic[63] profile and freely varying normalization in each filter (for example, refs. 65–67). Notably, by fitting to the individual exposures, ForcePho avoids correlated noise between pixels in drizzled mosaic images, enabling us to investigate scales smaller than individual pixels. The results are shown in Extended Data Fig. 1 and the resulting photometry is listed in Extended Data Table 1. From this analysis, we find a formal upper limit (84th percentile) on the half-light radius of 5.1 mas. We therefore conclude that the imaging data are consistent with the continuum source being unresolved. On the basis of tests with similarly faint brown dwarf stars that allow the expected systematic uncertainties to be quantified, we conservatively use an upper limit on the half-light radius as reported in ref. 16 for the F200W filter, $\lesssim$10 mas or 35 pc.

### Emission-line properties

The emission line at 1.71 μm is clearly and consistently detected across different PRISM data reductions, even when only one of the

two individual visits is considered (Supplementary information). We first fit a Gaussian profile to the sigma-clipped spectrum using the corresponding covariance matrix (Supplementary information), which provides a good fit to the data: $\chi^2 = 5.97$ with five degrees of freedom. We obtain a centroid of $1.7084 \pm 0.0014$ μm and FWHM = $302 \pm 18$ Å (or $\Delta v \approx 5{,}000$ km s$^{-1}$) that spans 2.4 wavelength bins (120 Å wide at 1.71 μm). We conclude that the line is probably unresolved in the PRISM spectrum and, as expected for compact sources observed with the NIRSpec MSA[68], that the spectral resolution is enhanced by a factor of approximately 1.5× compared with the resolution curve predicted for a uniformly illuminated microshutter.

To measure the absolute flux of the line, we first applied a correction to both the sigma-clipped spectrum and the covariance matrix based on the linear ForcePho fit found in our path-loss analysis (Supplementary information) to account for further path losses in the NIRSpec measurements. Directly integrating the corrected PRISM spectrum across the four wavelength bins between 1.69 μm and 1.73 μm (each bin with SNR > 1; Supplementary Information), we find a flux of $F = 7.42 \pm 1.16 \times 10^{-19}$ erg s$^{-1}$ cm$^{-2}$ (that is, the line is detected at SNR = 6.4). We have verified that all different data reductions (see Supplementary information) yield measurements consistent within 1$\sigma$. Specifically, the two visits independently confirm the line detection with measured fluxes of $5.77 \pm 1.36 \times 10^{-19}$ erg s$^{-1}$ cm$^{-2}$ and $9.07 \pm 1.80 \times 10^{-19}$ erg s$^{-1}$ cm$^{-2}$, respectively.

The emission line is not detected in the medium-resolution G140M/F070LP or G235M/F170LP spectra, both of which cover 1.71 μm, as shown in Extended Data Fig. 2 (although we note that the G235M/F170LP transmission drops below 1.7 μm; ref. 17). To quantify whether this is expected, taking into account their inherently lower sensitivity and relatively short exposure times compared with the PRISM spectra (NIRSpec observations and data reduction), we tested whether the observed R1000 spectra are consistent with the line flux measured in the PRISM spectra. Indeed, we find that, if the observed line profile is sufficiently broadened (FWHM $\gtrsim 600$ km s$^{-1}$, that is, well resolved at $R = 1{,}000$ resolution), it would be below the current sensitivity ($\lesssim 2\sigma$ detection expected; Extended Data Fig. 2).

As discussed further in the Supplementary information, we find it highly unlikely that the emission line at 1.71 μm is because of contamination of the microshutter by a foreground source that is aligned with JADES-GS-z13-1-LA by chance and remains undetected in the continuum, given that the continuum emission of JADES-GS-z13-1-LA unambiguously places the source at $z \approx 13$. We have performed the 'redshift sweep' analysis detailed in the appendices of refs. 5,9, in which the inferred one-sided $P$-value for a set of different emission lines is combined to yield the statistical significance of a potential spectroscopic confirmation at a given redshift. The effectiveness of this method is illustrated by the case of JADES-GS-z14-0, for which the most probable redshift was revealed[5] to be $z = 14.178$ (combined $P = 0.0072$), mainly based on a 3.6$\sigma$ detection of C III]. This redshift, consistent within the uncertainty determined from fitting the Ly-α break profile with DLA absorption, was later independently confirmed through the detection[21,69] of the [O III] 88 μm emission line by the Atacama Millimeter/submillimeter Array (ALMA). In the case of JADES-GS-z13-1-LA, the redshift sweep was performed across a range of $\Delta z = 0.2$ centred on $z = 13.0$, which, however, did not show any significant line detections.

Upper limits on the flux and EW for other, undetected, lines at $z = 13$ are therefore determined from integrating the covariance matrix across three PRISM wavelength bins, taking into account any residual flux after having subtracted a power-law model continuum (see 'Spectral modelling'). The resulting limits, summarized in Extended Data Table 2, are consistent with findings on most other $z > 10$ galaxies observed by the JWST, which have generally revealed these lines to be relatively weak[3,4,6,23,24].

## Spectral modelling

To gain insight into the Ly-α emission and absorption properties of JADES-GS-z13-1-LA, we model the observed spectrum with a simple framework in which Ly-α and continuum emission produced inside the central galaxy are subject to (damping-wing) absorption arising in intervening neutral hydrogen in dense absorbing systems and/or the IGM. We emphasize that the aim of this model is not to be as physically detailed as possible, which would involve performing simulations including three-dimensional radiative transfer coupled to the hydrodynamics of the gas (requiring the relevant feedback processes to be accurately modelled), but rather to constrain the basic physical properties that JADES-GS-z13-1-LA must have to explain the observations.

As we expect the Ly-α line to be redshifted with respect to the systemic redshift of the galaxy (potentially already as Ly-α emerges from the galaxy or otherwise resulting from processing by the neutral IGM[70,71]) and no other emission lines are detected (see 'Emission-line properties'), this quantity ($z_{sys}$) is not precisely known and is a free parameter in this model. To remain agnostic about the nature of the ionizing source and to avoid the intrinsic limitations of standard SPS models in reproducing very blue UV continua (Supplementary information), the continuum emission is modelled as a power law, $F_\lambda \propto \lambda^{\beta_{UV}}$, by default. This introduces two more free parameters in the model, the UV slope $\beta_{UV}$ and a normalization (at a rest-frame wavelength of $\lambda_{emit} = 1{,}500$ Å).

To reproduce the smooth Ly-α break seen in the continuum, we allow the continuum emission to be affected by DLA absorption parameterized by the neutral hydrogen column density $N_{HI}$ as in refs. 6,9. The Ly-α emission is explicitly not attenuated by this absorption, as this would completely extinguish the line. Because the attenuated continuum tends to zero at the wavelength of Ly-α, we calculate the line EW according to the unattenuated continuum, which is effectively equivalent to measuring the continuum level by means of the photometry. As discussed in the main text, this would require a specific geometrical configuration such that the Ly-α emission is not strongly absorbed. However, Ly-α emission superimposed on DLA troughs has been observed in galaxy spectra, suggesting that these geometries exist[45,46]. The absorption cross-section of neutral hydrogen is based on the Voigt profile approximation given by in ref. 72, with a quantum-mechanical correction provided in ref. 73. Because we find that the redshift of the foreground DLA system (when freely varied; for example, ref. 74) prefers a solution close to the systemic redshift, $z_{DLA} \approx z_{sys}$, for simplicity, we fix $z_{DLA} = z_{sys}$ in the following.

Alternatively, we considered the case in which the observed spectrum is dominated by the 2γ continuum, which has a fixed shape[75] and thus only requires one free parameter, the normalization. As a third variant, we considered a combination consisting of a power-law continuum (using the same parameterization as above) and a full nebular emission spectrum, which, as well as the 2γ continuum and the Ly-α line, also contains the free-bound and free-free components. The nebular emission in this case was computed with the PyNeb code[76], which, however, requires assuming the gas temperature and density. We opted for $T = 20{,}000$ K and $n = 100$ cm$^{-3}$, respectively, for which the 2γ continuum is the dominant contributor in the wavelength range considered here[77]. The choice for this relatively low density is motivated by the fact that the free-bound (and free-free) components mainly contribute at longer wavelengths and would have to be subdominant to reproduce the very steep UV slope. In this multicomponent ('self-consistent') model, we tied the continuum normalization to the strength of the Ly-α line, thereby self-consistently scaling the continuum according to the production rate and escape fraction of LyC photons discussed below.

Following refs. 11,78, IGM transmission was calculated with the patchy reionization model presented in ref. 79, integrating along the trajectory of a photon that starts in an ionized bubble of radius $R_{ion}$ located in an otherwise neutral IGM (see also refs. 80,81). Following ref. 79, we assume the gas in the ionized bubble to be highly ionized (residual neutral fraction fixed at $x_{HI} = 10^{-8}$) and have $T = 10^4$ K, whereas the neutral IGM is at $T = 1$ K. The gas in both media is assumed to be at mean cosmic density (that is, to have $\bar{n}_H \approx 5.25 \times 10^{-4}$ cm$^{-3}$ at $z = 13$) and be

at rest with respect to the central source. We fixed the global neutral hydrogen fraction of the IGM (that is, outside the ionized bubble[78]) to $\bar{x}_{HI} = 1$, motivated by various types of evidence that consistently indicate that, globally, the Universe is still highly neutral well below redshift $z = 13$ (for example, refs. 82,83).

We self-consistently model the size of the ionized bubble by considering the production rate and escape fraction of hydrogen-ionizing photons of the central galaxy. As in ref. 11, we define $\xi_{ion} \equiv \dot{N}_{ion}/L_{\nu,UV}$, in which $\dot{N}_{ion}$ is the production rate of ionizing photons and $L_{\nu,UV}$ is the luminosity density (in units of erg s$^{-1}$ Hz$^{-1}$) of the intrinsic continuum of the ionizing source at $\lambda_{emit} = 1,500$ Å. In the case of the multicomponent model in particular, $L_{\nu,UV}$ is taken to be the value of the power-law continuum at 1,500 Å such that $\xi_{ion}$ reflects the intrinsic value. The rate of ionizing photons leaking from the galaxy at a given production efficiency $\xi_{ion}$ is modulated by the LyC escape fraction, $f_{esc,LyC}$. In a given model instance, we therefore begin by deriving the rate of ionizing photons escaping the galaxy using (for example, refs. 26,84–86)

$$\dot{N}_{ion,esc} = f_{esc,LyC} \dot{N}_{ion} = f_{esc,LyC} \xi_{ion} L_{\nu,UV}. \qquad (1)$$

To calculate the bubble radius $R_{ion}$, we then numerically integrate equation (3) in ref. 80, describing the time evolution of $R_{ion}(t)$ to obey

$$\frac{dR_{ion}^3}{dt} = 3H(z)R_{ion}^3 + \frac{3\dot{N}_{ion,esc}}{4\pi\bar{n}_H} - C_{HII}\bar{n}_H\alpha_B R_{ion}^3, \qquad (2)$$

thereby taking into account the effect of the expansion of the Universe parameterized by the Hubble parameter $H(z)$ and recombinations within the ionized bubble, for which we assume a clumping factor for ionized gas of $C_{HII} = 3$ (for example, ref. 87) and case B recombination rate $\alpha_B$ at 20,000 K, as given by in ref. 88. The typical recombination timescale at $z = 13$, $(C_{HII}\bar{n}_H\alpha_B)^{-1} \approx 140$ Myr, indicates that JADES-GS-z13-1-LA as an ionizing source could quickly ionize its surroundings before recombinations are able to restore balance. As illustrated in Extended Data Fig. 4, showing the time evolution of $R_{ion}$ in the default model, the bubble radius can reach $R_{ion} \approx 0.1$ pMpc over a timescale of only 1 Myr. We note that, when the supply of LyC photons stops, the residual neutral hydrogen fraction rapidly increases owing to the high density at $z = 13$ ($x_{HI} \approx 0.01$ after 1 Myr), implying that, for an ionized bubble to have a substantial transmission-enhancing effect redwards of the systemic Ly-α wavelength[79], it must be actively maintained. Here we integrate until we reach a fiducial age of $t = 10$ Myr, having verified that changing this assumption has little impact on our findings as a result of the sublinear scaling $R_{ion} \propto t^{1/3}$ (in the absence of recombinations and the Hubble flow). We also considered an alternative model identical to the default power-law model but for which we fix $R_{ion} = 0$ (that is, $f_{esc,LyC} = 0$).

Finally, we determine the intrinsic Ly-α luminosity resulting from recombinations by considering the number of ionizing photons that are absorbed within the galaxy and reprocessed into Ly-α. Similarly to the above, the effective rate of LyC photons contributing to the recombination rate within the galaxy ($\dot{N}_{rec}$) follows from the product of the intrinsic production rate $\dot{N}_{ion}$ and absorbed fraction (one minus the escape fraction) of ionizing photons. This is multiplied by the fraction of (case B) recombination events that result in the emission of a Ly-α photon, $f_{rec,B}$ (see, for example, ref. 89), to arrive at the emission rate of Ly-α photons and hence the Ly-α luminosity (that is, the emission rate multiplied by the energy of a Ly-α photon),

$$\begin{aligned} L_{Ly-\alpha} &= \dot{N}_{rec}f_{rec,B}E_{Ly-\alpha} \\ &= (1-f_{esc,LyC})\xi_{ion}L_{\nu,UV}f_{rec,B}E_{Ly-\alpha}. \end{aligned} \qquad (3)$$

We used $f_{rec,B}(T = 20,000 \text{ K}) = 0.647$ based on the prescription in ref. 90, noting that it depends only weakly on temperature[91] and that case A would lead to an unaccounted increase in $f_{esc,LyC}$. Under the very

conservative assumptions of no IGM absorption at all and $f_{esc,LyC} = 0$, equation (3) places a lower limit on the LyC production efficiency through the observed Ly-α luminosity relative to the continuum, yielding $\xi_{ion} \gtrsim 10^{25.1}$ Hz erg$^{-1}$ ($10^{25.4}$ Hz erg$^{-1}$ under case A). The Ly-α line, as it emerges from the galaxy, is modelled as a Gaussian profile with a given velocity dispersion $\sigma_{Ly-\alpha}$, which is shifted in velocity space at a given offset from the systemic redshift, $\Delta\nu_{Ly-\alpha,int}$, and normalized to the Ly-α luminosity derived as described above.

Radiative transfer calculations predict that a wide variety of Ly-α spectral profiles may emerge from galaxies[92,93], but galactic outflows typically cause systematically redshifted components[94,95], as seen ubiquitously at high redshift[70,71,96–99]. Although the emergent Ly-α spectral profile is fundamentally unknown at $z \gtrsim 7$ owing to the asymmetric IGM transmission on the blue side[78,79], some clues are given by the non-detection of the line in the R1000 spectra. If the line were unresolved at a resolution of $R \approx 1000$, that is FWHM $\lesssim 300$ km s$^{-1}$, we would have probably seen a marginal detection (Extended Data Fig. 2). Instead, the line profile probably contains a prominent red, broad component to allow for sufficient transmission of Ly-α flux at $z = 13$ even in the presence of an ionized bubble (Fig. 2). We note that, because of the IGM transmission, the peak of the intrinsic line profile at velocity offset $\Delta\nu_{Ly-\alpha,int}$ with respect to systemic redshift effectively gets further redshifted to a velocity offset of $\Delta\nu_{Ly-\alpha,obs}$.

We used the PyMultiNest[100] implementation of the multimodal nested-sampling algorithm MultiNest[101] to perform a Bayesian fitting routine to the sigma-clipped PRISM spectrum and corresponding covariance matrix (see Supplementary information) from 1.609 μm up to 2.897 μm (127 wavelength bins), or 1,150 Å $\lesssim \lambda_{emit} \lesssim 2,000$ Å at $z = 13$. Before fitting, as in the section 'Emission-line properties', we corrected the NIRSpec measurements for further path losses. Meanwhile, the model spectrum is convolved with the PRISM resolution curve predicted for a uniformly illuminated microshutter, enhanced by a factor of 1.5 based on the measured width of the Ly-α line in the PRISM spectrum (see 'Emission-line properties'). As detailed by Jakobsen et al. (manuscript in preparation), the goodness-of-fit statistic $\chi^2$ is calculated as the matrix product

$$\chi^2 = \mathbf{R}^T\mathbf{\Sigma}^{-1}\mathbf{R}, \qquad (4)$$

in which $\mathbf{\Sigma}^{-1}$ is the inverted covariance matrix and the $i$th element of the vector $\mathbf{R}$ is given as the difference between observed flux density in the $i$th wavelength bin ($F_{\lambda,i}^{obs}$) and the modelled one ($F_{\lambda,i}^{model}$),

$$R_i = F_{\lambda,i}^{obs} - F_{\lambda,i}^{model}. \qquad (5)$$

The model log-likelihood $\ell$ is calculated assuming that the observed data are normally distributed around the model, $\ell = -\frac{1}{2}\chi^2$. All model parameters, prior distributions and resulting best-fitting values are summarized in Extended Data Table 3. The posterior distributions for the default model are shown in Extended Data Fig. 3.

Although the multicomponent self-consistent model has a slightly higher $\chi^2$ (171.4) than the default power-law model ($\chi^2 = 168.1$), notably, it favours a high LyC escape fraction ($f_{esc,LyC} = 0.73^{+0.14}_{-0.26}$) to suppress the nebular continuum, much like the SPS model fits (Supplementary information). Indeed, fixing $R_{ion} = 0$ in the self-consistent model (results not included here) yields a much poorer fit ($\chi^2 = 183.1$), as this overpredicts the continuum tied to the strong Ly-α line. Moreover, the intrinsic Ly-α flux required for the $R_{ion} = 0$ power-law model is discrepant at a $4.5\sigma$ level with the MIRI/F770W non-detection (see 'Photometric measurements').

## Data availability

The NIRCam data that support the findings of this study are publicly available at https://archive.stsci.edu/hlsp/jades. The reduced spectra that support the findings of this study are publicly available at https://doi.org/10.5281/zenodo.14714293 (ref. 102).

## Code availability

The code used for the spectral modelling fitting routine is available at https://github.com/joriswitstok/lymana_absorption. The Astropy[103,104] software suite is publicly available, as is BAGPIPES[105], BEAGLE[106], CLOUDY[107], emcee[108], ForcePho[109], MultiNest[101], PyMultiNest[100], PyNeb[76], the SciPy library[110], its packages NumPy[111] and Matplotlib[112], and SpectRes[113].

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

**Acknowledgements** We thank L. Keating, H. Katz, C. Witten, W. McClymont, A. van der Wel, J. Chisholm, D. Berg and M. Ouchi for useful discussions. This work is based on observations made with the National Aeronautics and Space Administration (NASA)/European Space Agency (ESA)/Canadian Space Agency (CSA) JWST. The data were obtained from the Mikulski Archive for Space Telescopes at the STScI, which is operated by the Association of Universities for Research in Astronomy, Inc., under NASA contract NAS 5-03127 for the JWST. These observations are associated with programmes 1180, 1210, 1286, 1287 and 3215. J.W., R.M., W.M.B., F.D. and J.S. acknowledge support from the Science and Technology Facilities Council (STFC), by the European Research Council (ERC) through Advanced Grant 695671 'QUENCH' and by the UK Research and Innovation (UKRI) Frontier Research grant RISEandFALL. J.W. also gratefully acknowledges support from the Cosmic Dawn Center through the DAWN Fellowship. The Cosmic Dawn Center is financed by the Danish National Research Foundation under grant no. 140. B.D.J., B.E.R., F.S., P.A.C., D.J.E., C.N.A.W. and Y.Z. acknowledge support from the JWST/NIRCam contract to the University of Arizona, NAS5-02015. B.E.R. also acknowledges support from JWST Program 3215. S.T. acknowledges support from Royal Society Research Grant G125142. A.J.C., A.J.B., A.S., J.C. and G.C.J. acknowledge support from the 'FirstGalaxies' Advanced Grant from the ERC under the European Union's Horizon 2020 research and innovation programme (grant agreement no. 789056). R.S. acknowledges support from a STFC Ernest Rutherford Fellowship (ST/S004831/1). S. Alberts acknowledges support from the JWST MIRI Science Team Lead, grant 80NSSC18K0555 and from NASA Goddard Space Flight Center to the University of Arizona. S. Arribas acknowledges grant PID2021-127718NB-I00 financed by the Spanish Ministry of Science and Innovation/State Agency of Research (MICIN/AEI/10.13039/501100011033). This research is supported in part by the Australian Research Council Centre of Excellence for All Sky Astrophysics in 3 Dimensions (ASTRO 3D), through project number CE170100013. S. Carniani acknowledges support by the European Union's HE ERC Starting Grant No. 101040227 'WINGS'. E.C.-L. acknowledges the support of an STFC Webb Fellowship (ST/W001438/1). D.J.E. is supported as a Simons Investigator.

P.G.P.-G. acknowledges support from grant PID2022-139567NB-I00 financed by Spanish Ministerio de Ciencia e Innovación MCIN/AEI/10.13039/501100011033, FEDER, UE. H.Ü. gratefully acknowledges support by the Isaac Newton Trust and by the Kavli Foundation through a Newton-Kavli Junior Fellowship. H.Ü. also acknowledges funding by the European Union (ERC APEX, 101164796). Views and opinions expressed are however those of the authors only and do not necessarily reflect those of the European Union or the European Research Council Executive Agency. Neither the European Union nor the granting authority can be held responsible for them. The research of C.C.W. is supported by NOIRLab, which is managed by the Association of Universities for Research in Astronomy (AURA) under a cooperative agreement with the National Science Foundation. This study made use of the Prospero high-performance computing facility at Liverpool John Moores University.

**Author contributions** J.W. and P.J. led the analysis and the writing of the paper, with key contributions from A.J.B., A.J.C., A.S., B.D.J., B.E.R., F.S., J.M.H., M.C., R.M., R.S., S. Carniani and S.T. A.J.B., C.W., F.D., G.C.J., J.C., J.W., K.B., M.C., N.K., P.J., R.M., S. Arribas, S. Carniani and S. Charlot contributed to the development and commissioning of the NIRSpec instrument and the reduction and analysis of the NIRSpec data presented. B.D.J., B.E.R., C.C.W., C.N.A.W., D.J.E., F.S., K.N.H., P.A.C. and S.T. contributed to the development and commissioning of the NIRCam instrument and the reduction and analysis of the NIRCam data presented. J.M.H. and S. Alberts contributed to the reduction and analysis of the MIRI data presented. A.J.B., B.D.J., B.E.R., B.D.J., and R.M. contributed to the design and execution of the JADES programme. A.J.B., B.E.R., C.W., D.J.E. and S.T. serve on the JADES Steering Committee. R.B., W.M.B., P.R. and Y.Z. provided comments on the manuscript.

**Competing interests** The authors declare no competing interests.

## Additional information
**Correspondence and requests for materials** should be addressed to Joris Witstok.

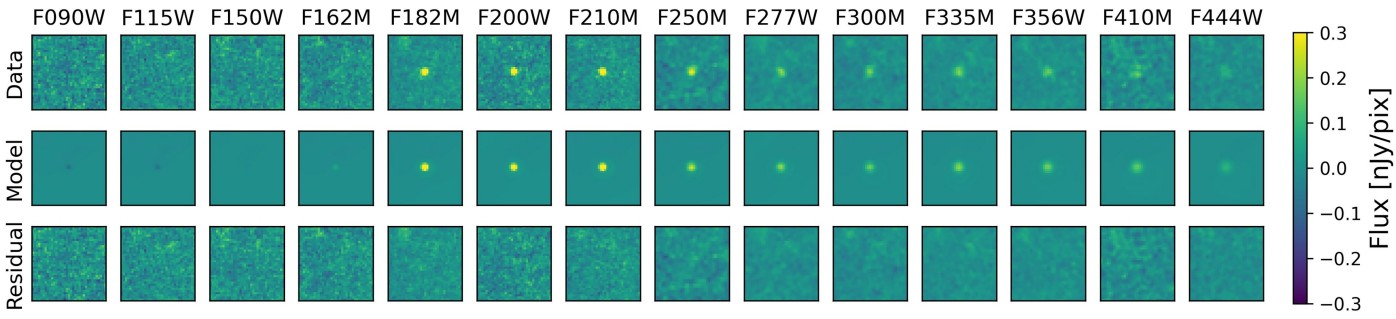

**Extended Data Fig. 1 | ForcePho modelling of JADES-GS-z13-1-LA.** The top row shows roughly 1″ × 1″ cutouts of the observed data (scaled according to the colour bar shown on the right) around JADES-GS-z13-1-LA in each of the 14 available NIRCam filters, as annotated at the top of each column. The PSF-convolved ForcePho model (see 'Photometric measurements') is shown in the middle row. The bottom row shows that residuals between data and model are consistent with pure noise, indicating that the model provides a good fit to the data. Note that, although the ForcePho fits are performed on more than 400 separate exposures, they are mosaiced together here to visualize the data and residuals.

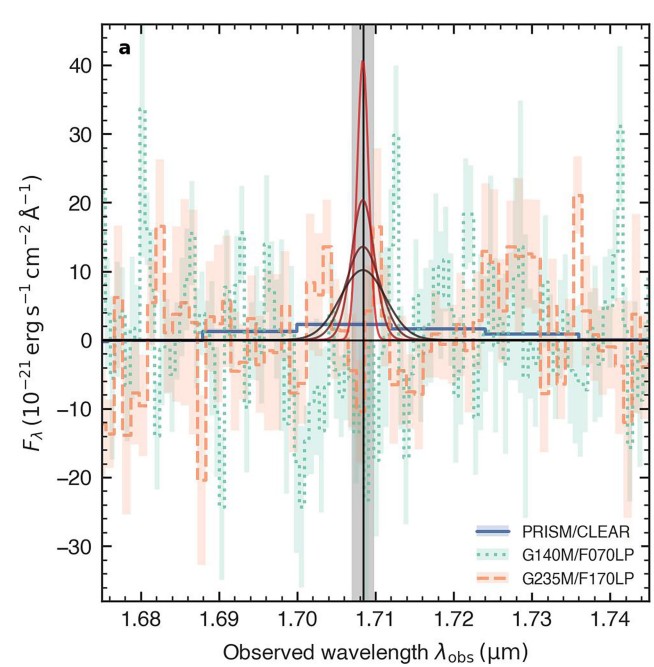

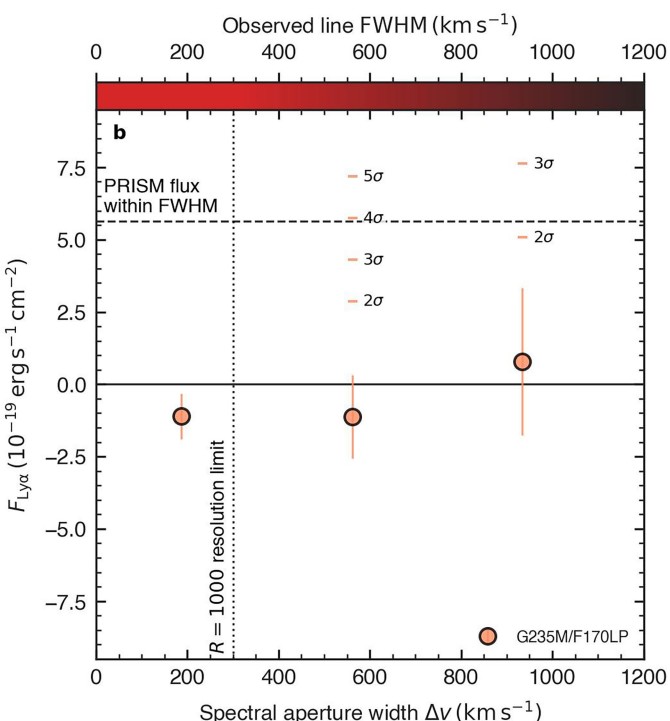

**Extended Data Fig. 2 | Medium-resolution (R1000) grating spectra of JADES-GS-z13-1-LA. a**, Coloured lines represent observed spectra in different grating-filter modes, as obtained from the sigma-clipping procedure (Supplementary information). Specifically, we show the G140M/F070LP (dotted turquoise line) and G235M/F170LP (dashed range line) spectra compared with the low-resolution PRISM spectrum (dark blue line). Shading represents a $1\sigma$ uncertainty on all components of the figure. Solid curves represent emission-line profiles at increasing widths (according to the colour bar in panel **b**), starting from the $R = 1,000$ resolution limit and having matched the flux and central wavelength (1.708 μm; indicated by a vertical black line) to

the values measured from the PRISM spectrum (see 'Emission-line properties'). **b**, Measured Ly-α flux in an increasingly wide spectral aperture centred on 1.708 μm in G235M/F170LP are shown by circles with $1\sigma$ error bars, none of which show a significant detection. This is consistent with the less sensitive G140M/F070LP measurements (not shown here for clarity). A horizontal dashed line shows the measured PRISM line flux contained within the FWHM of a Gaussian profile (76%), whereas a vertical dotted line indicates the limiting $R = 1,000$ resolution. This illustrates that, if the emission line is well resolved (FWHM ≳ 600 km s$^{-1}$), it would fall below the nominal noise level of the R1000 gratings (see annotated $2\sigma$ and $4\sigma$ levels).

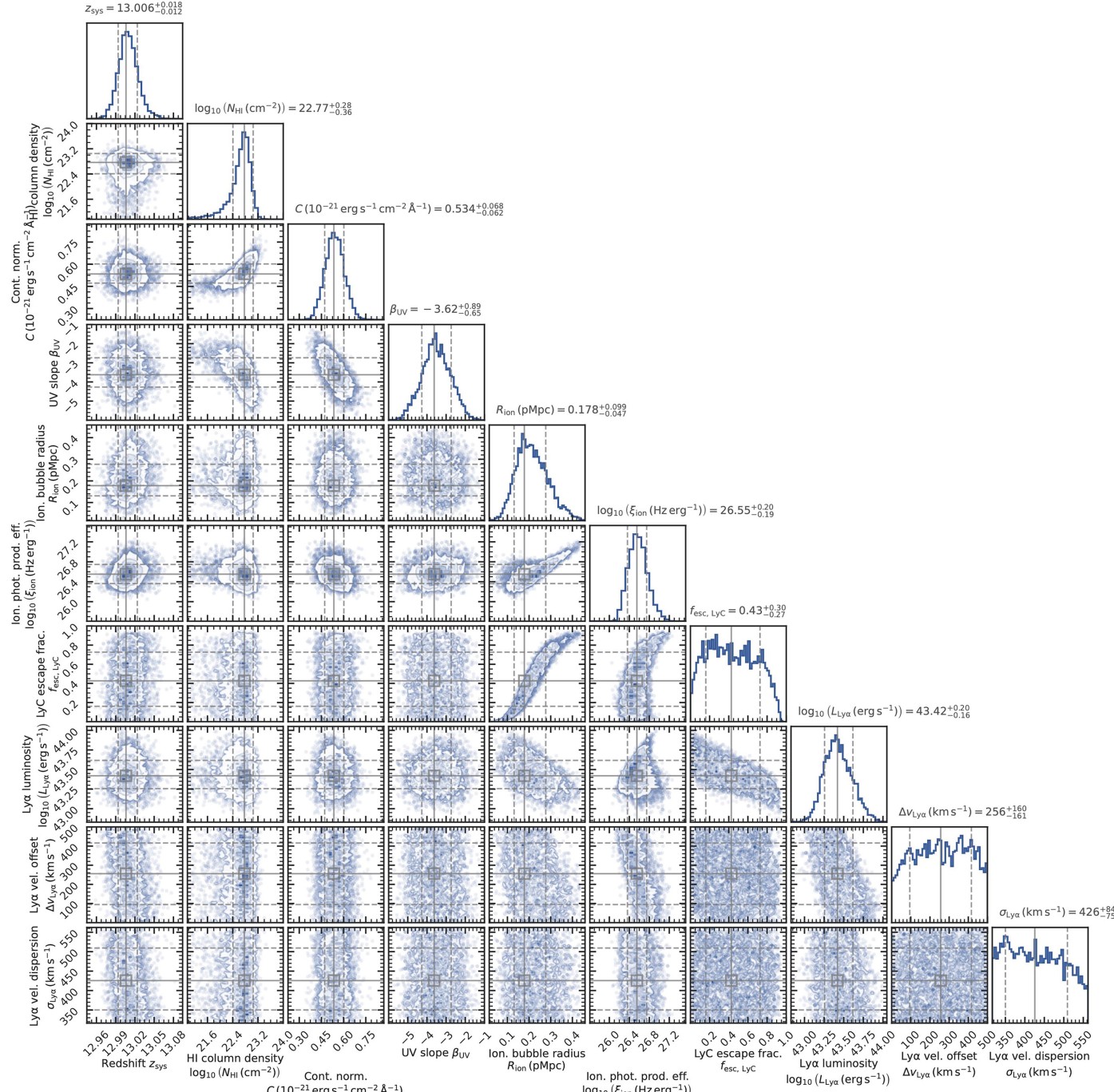

**Extended Data Fig. 3 | Posterior distributions from spectral modelling of the observed spectrum of JADES-GS-z13-1-LA.** The small panels show inter-dependencies between all eight parameters freely varied in the model (Extended Data Table 3). Furthermore, we include the physical radius of the ionized bubble ($R_{ion}$) and Ly-α luminosity ($L_{Ly\text{-}\alpha}$), which are not independently varied but are instead determined by the other parameters (see 'Spectral modelling').

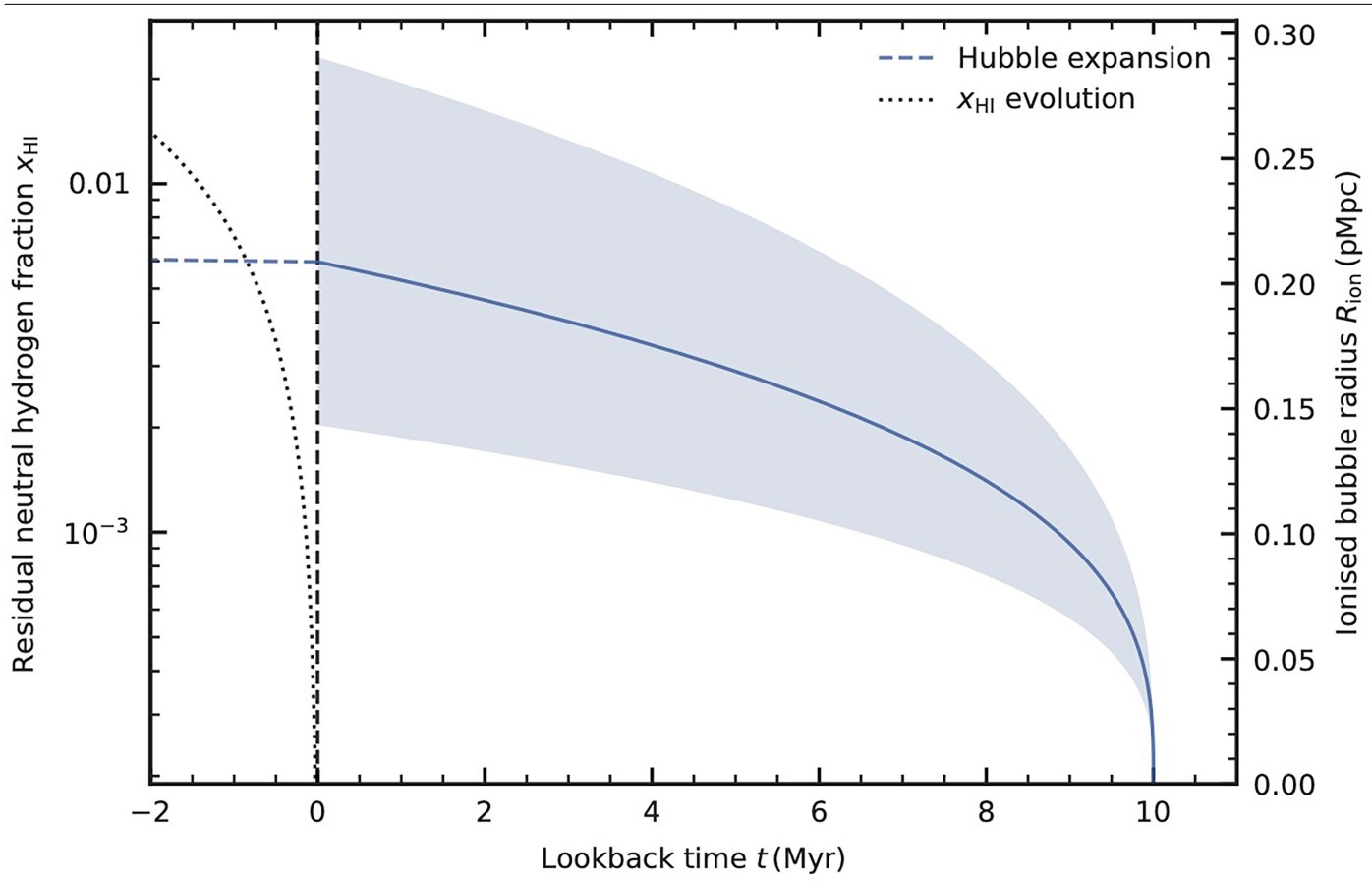

**Extended Data Fig. 4 | Modelled ionized bubble size evolution.** The right axis shows the physical radius of the ionized bubble $R_{ion}$, whose evolution as a function of lookback time $t$ is governed by equation (2). The solid line shows the median among the posterior distribution of the default model and the shading represents $1\sigma$ uncertainty (16th to 84th percentile). The dashed line illustrates the Hubble expansion rate if the bubble remains unchanged from $t = 0$ onwards, showing that this effect has little impact over the timescale relevant to our analysis. The dotted line shows how the neutral hydrogen fraction within the bubble (left axis) would evolve without further ionizing photons.

**Extended Data Table 1 | Photometry and UV-continuum properties of JADES-GS-z13-1-LA**

| Quantity | Instrument | Filter (set) | ForcePho | CIRC2 | Synthetic (NIRSpec) |
|---|---|---|---|---|---|
| | HST/ACS | F435W | . . . | $2.37 \pm 3.81$ | . . . |
| | | F606W | . . . | $-0.80 \pm 3.02$ | . . . |
| | | F775W | . . . | $-3.00 \pm 4.34$ | . . . |
| | | F814W | . . . | $6.11 \pm 3.79$ | . . . |
| | | F850LP | . . . | $-2.25 \pm 7.35$ | . . . |
| | HST/WFC3 | F125W | . . . | $-34.6 \pm 15.9$ | . . . |
| | | F160W | . . . | $-12.1 \pm 22.0$ | . . . |
| | JWST/NIRCam | F090W | $-0.448 \pm 0.190$ | $-0.372 \pm 0.609$ | $-0.023 \pm 0.641$ |
| | | F115W | $-0.157 \pm 0.163$ | $0.664 \pm 0.496$ | $-0.354 \pm 0.551$ |
| | | F150W | $0.023 \pm 0.183$ | $-0.430 \pm 0.480$ | $0.013 \pm 0.515$ |
| | | F162M | $0.696 \pm 0.211$ | $1.270 \pm 0.566$ | $1.212 \pm 0.780$ |
| Flux density $F_\nu$ (nJy) | | F182M | $7.164 \pm 0.201$ | $8.097 \pm 0.331$ | $4.082 \pm 0.701$ |
| | | F200W | $6.248 \pm 0.267$ | $5.718 \pm 0.488$ | $4.632 \pm 0.496$ |
| | | F210M | $7.025 \pm 0.245$ | $8.077 \pm 0.416$ | $4.973 \pm 0.712$ |
| | | F250M | $5.421 \pm 0.397$ | $6.941 \pm 0.211$ | $3.857 \pm 0.998$ |
| | | F277W | $4.624 \pm 0.259$ | $5.101 \pm 0.183$ | $3.309 \pm 0.511$ |
| | | F300M | $4.061 \pm 0.231$ | $4.569 \pm 0.166$ | $2.835 \pm 0.729$ |
| | | F335M | $5.304 \pm 0.310$ | $5.951 \pm 0.173$ | $3.582 \pm 0.821$ |
| | | F356W | $5.008 \pm 0.296$ | $5.514 \pm 0.199$ | $3.787 \pm 0.644$ |
| | | F410M | $4.883 \pm 0.513$ | $4.400 \pm 0.320$ | $3.19 \pm 1.10$ |
| | | F444W | $3.890 \pm 0.433$ | $3.692 \pm 0.273$ | $1.488 \pm 0.942$ |
| | JWST/MIRI | F770W | | $1.60 \pm 2.23$ | |
| UV magnitude $M_{\mathrm{UV}}$ (mag) | | F210M through F444W | $-18.492^{+0.039}_{-0.038}$ | $-18.601^{+0.036}_{-0.035}$ | $-18.03^{+0.15}_{-0.13}$ |
| Bolometric luminosity $L_{\mathrm{bol}}$ ($10^{10}$ L$_\odot$) | | F210M through F444W | $9.0^{+1.6}_{-1.4}$ | $9.6^{+1.4}_{-1.2}$ | $7.8^{+6.6}_{-3.5}$ |
| UV slope (up to 5 μm) $\beta_{\mathrm{UV}}$ | | F210M through F444W | $-2.75^{+0.10}_{-0.10}$ | $-2.719^{+0.082}_{-0.082}$ | $-2.88^{+0.36}_{-0.37}$ |
| UV slope (up to 3.5 μm) $\hat{\beta}_{\mathrm{UV}}$ | | F210M through F335M | $-3.10^{+0.14}_{-0.14}$ | $-2.81^{+0.13}_{-0.13}$ | $-3.19^{+0.57}_{-0.56}$ |

Reported quantities (and corresponding 1σ uncertainties) are the flux density $F_\nu$ in nJy, the UV magnitude ($M_{\mathrm{UV}}$) in magnitudes, the bolometric luminosity ($L_{\mathrm{bol}}$) in $10^{10}$ Solar luminosity and UV slopes taking into account all available filters redwards of 2 μm ($\beta_{\mathrm{UV}}$) or only up to and including F335M ($\hat{\beta}_{\mathrm{UV}}$). Fluxes in available HST and JWST filters are measured with ForcePho and within circular 0.3″-diameter apertures (CIRC2), except for MIRI/F770W, as detailed in 'Photometric measurements'. Synthetic photometry in NIRCam filters is directly extracted from the NIRSpec/PRISM spectrum (see Supplementary information). For each of the three different sets of photometry, UV properties ($L_{\mathrm{bol}}$, $M_{\mathrm{UV}}$ and $\beta_{\mathrm{UV}}$) are measured redwards of $\lambda_{\mathrm{obs}} = 2.0$ μm, corresponding to rest-frame wavelengths $\lambda_{\mathrm{emit}} \gtrsim 1,500$ Å at $z = 13$ (see Supplementary information for details). The uncertainty on the UV magnitude ($M_{\mathrm{UV}}$) takes into account a systematic uncertainty of $\Delta z = 0.05$.

**Extended Data Table 2 | Emission-line constraints for JADES-GS-z13-1-LA**

| Emission line(s) | $F$ ($10^{-19}$ erg s$^{-1}$ cm$^{-2}$) | EW (Å) |
|---|---|---|
| Lyα | $7.42 \pm 1.16$ | $> 40^*$ |
| He II $\lambda$ 1640 Å | $< 1.6$ | $< 30$ |
| N IV] $\lambda$ 1483, 1487 Å | $< 2.0$ | $< 29$ |
| C IV $\lambda$ 1548, 1551 Å | $< 1.6$ | $< 27$ |
| [O III] $\lambda$ 1660, 1666 Å | $< 1.2$ | $< 25$ |
| N III] | $< 1.8$ | $< 43$ |
| C III] | $< 1.1$ | $< 36$ |
| [O II] $\lambda$ 3727, 3730 Å | $< 0.95$ | $< 42$ |

Presented quantities for each line are the flux and EW from the PRISM spectra. Constraints for undetected lines are presented as 3σ upper limits. N III] refers to the multiplet at 1,750 Å, whereas C III] is shorthand for [C III]$\lambda$1,907 Å, C III]$\lambda$1,909 Å.

*Discussed in more detail in 'Spectral modelling'.

# Extended Data Table 3 | Spectral model parameters, prior distributions and best-fitting values

| Parameter | (Logarithmic) unit | Type | Prior | Min. | Max. | Default (power law) | Pure $2\gamma$ | Self-consistent | Fixed $R_{\rm ion}=0$ |
|---|---|---|---|---|---|---|---|---|---|
| $z_{\rm sys}$ | | Varied | Uniform | 12.85 | 13.1 | $13.01^{+0.02}_{-0.01}$ | $13.01^{+0.01}_{-0.02}$ | $13.01^{+0.02}_{-0.02}$ | $12.99^{+0.02}_{-0.02}$ |
| $\log_{10}(N_{\rm H\,I})$ | $\rm cm^{-2}$ | Varied | Uniform | 19 | 24 | $22.77^{+0.28}_{-0.36}$ | $19.58^{+1.10}_{-0.41}$ | $22.60^{+0.51}_{-1.24}$ | $22.79^{+0.31}_{-0.40}$ |
| $C$ | $10^{-21}\,\rm erg\,s^{-1}\,cm^{-2}\,\mathring{A}^{-1}$ | Varied | Uniform | 0 | 1 | $0.534^{+0.068}_{-0.062}$ | $0.444^{+0.031}_{-0.030}$ | $0.215^{+0.104}_{-0.056}$ * | $0.546^{+0.063}_{-0.081}$ |
| $\beta_{\rm UV}$ | | Varied | Uniform | $-6$ | $-1$ | $-3.62^{+0.89}_{-0.65}$ | $-$ | $-5.52^{+1.19}_{-0.43}$ | $-3.60^{+0.93}_{-0.72}$ |
| $R_{\rm ion}$ | pMpc | Coupled/fixed | $-$ | $-$ | $-$ | $0.178^{+0.099}_{-0.047}$ | $0.177^{+0.102}_{-0.057}$ | $0.253^{+0.092}_{-0.074}$ | $0^\dagger$ |
| $\log_{10}(\xi_{\rm ion})$ | $\rm Hz\,erg^{-1}$ | Varied | Uniform | 24 | 28 | $26.55^{+0.20}_{-0.22}$ | $26.62^{+0.17}_{-0.22}$ | $27.00^{+0.23}_{-0.32}$ * | $26.60^{+0.09}_{-0.14}$ |
| $f_{\rm esc,\,LyC}$ | | Varied/fixed | Uniform | 0 | 1 | $0.43^{+0.30}_{-0.27}$ | $0.42^{+0.34}_{-0.26}$ | $0.73^{+0.14}_{-0.26}$ | $0^\dagger$ |
| $L_{\rm Ly\alpha}$ | $10^{43}\,\rm erg\,s^{-1}$ | Coupled | $-$ | $-$ | $-$ | $2.6^{+1.5}_{-0.8}$ | $2.6^{+1.5}_{-0.9}$ | $1.6^{+0.3}_{-0.3}$ | $5.5^{+1.9}_{-0.9}$ |
| $\rm EW_{Ly\alpha,\,intr}$ | $\mathring{A}$ | Coupled | $-$ | $-$ | $-$ | $691^{+467}_{-254}$ | $-^\ddagger$ | $704^{+541}_{-289}$ | $1375^{+591}_{-420}$ |
| $\rm EW_{Ly\alpha,\,obs}$ | $\mathring{A}$ | Coupled | $-$ | $-$ | $-$ | $43^{+15}_{-11}$ | $-^\ddagger$ | $67^{+43}_{-23}$ | $41^{+16}_{-11}$ |
| $f_{\rm esc,\,Ly\alpha}$ | | Coupled | $-$ | $-$ | $-$ | $0.063^{+0.026}_{-0.023}$ | $0.063^{+0.027}_{-0.022}$ | $0.095^{+0.025}_{-0.020}$ | $0.030^{+0.008}_{-0.007}$ |
| $\Delta v_{\rm Ly\alpha,\,int}$ | $\rm km\,s^{-1}$ | Varied | Uniform | 0 | 500 | $256^{+160}_{-161}$ | $257^{+161}_{-163}$ | $379^{+83}_{-150}$ | $391^{+74}_{-111}$ |
| $\Delta v_{\rm Ly\alpha,\,obs}$ | $\rm km\,s^{-1}$ | Coupled | $-$ | $-$ | $-$ | $694^{+299}_{-299}$ | $603^{+299}_{-269}$ | $638^{+359}_{-299}$ | $930^{+329}_{-299}$ |
| $\sigma_{\rm Ly\alpha}$ | $\rm km\,s^{-1}$ | Varied | Log-uniform | $10^{2.5}$ | $10^{2.75}$ | $426^{+84}_{-75}$ | $430^{+82}_{-77}$ | $436^{+83}_{-81}$ | $457^{+73}_{-86}$ |
| $\chi^2$ | | | | | | 168.1 | 175.6 | 171.4 | 168.4 |
| DOF | | | | | | 119 | 120 | 119 | 120 |

Model parameters are the systemic redshift ($z_{\rm sys}$), DLA neutral hydrogen column density ($N_{\rm H\,I}$), power-law continuum normalization ($C$) and slope ($\beta_{\rm UV}$), ionizing-photon production efficiency ($\xi_{\rm ion}$) and escape fraction ($f_{\rm esc,LyC}$), and the peak velocity offset ($\Delta v_{\rm Ly\text{-}\alpha,int}$) and intrinsic velocity dispersion ($\sigma_{\rm Ly\text{-}\alpha}$) of Ly-$\alpha$ line profile emerging from the galaxy. Further reported parameters are the ionized bubble radius ($R_{\rm ion}$), Ly-$\alpha$ luminosity ($L_{\rm Ly\text{-}\alpha}$), Ly-$\alpha$ EW as it emerges from the galaxy ($\rm EW_{Ly\text{-}\alpha,intr}$) and as it is observed (that is, after IGM transmission; $\rm EW_{Ly\text{-}\alpha,obs}$), Ly-$\alpha$ escape fraction ($f_{\rm esc,Ly\text{-}\alpha}$) and the observed Ly-$\alpha$ velocity offset ($\Delta v_{\rm Ly\text{-}\alpha,obs}$), which are not freely varied but derived from the main parameters. Best-fitting values (uncertainties) are the median (16th and 84th percentiles) of the posterior distribution under the default (power-law) model, a model with pure two-photon continuum ($2\gamma$), a self-consistent model incorporating power-law and nebular-emission components and a power-law model in which $R_{\rm ion}=0$ (for details, see 'Spectral modelling').

*(Relative to the) power-law continuum only.

$\dagger$Value is fixed in this model.

$\ddagger$The $2\gamma$ continuum tends to zero approaching the wavelength of Ly-$\alpha$.