## [Peer Review File · Nature]

Witnessing the onset of Reionisation via Lyman- α emission at redshift 13

Corresponding Author: Dr Joris Witstok

Version 1:

Reviewer comments:

Referee #1

(Remarks to the Author)

Referee report for manuscript 485218 - Witstok et al.

The manuscript presents deep low-resolution JWST NIRSpec observations of a photometric candidate galaxy at redshift $z \sim 13$, with the analysis showing a clear continuum break in the spectrum and - remarkably - an emission line bluewards of the spectral break, which is identified as Ly α . The results presented are unique since none of the handful of galaxies spectroscopically confirmed at $z > 10$ have shown Ly α emission, as naturally expected because of the largely ionized nature of the intergalactic medium at those early times. The Ly α emission peak detected in this source appears to fall substantially redwards of the systemic galaxy redshift (velocity offset greater than 500 km/s). Also, the observations are at low spectral resolution yet there is some evidence presented to support a broadened line profile, with velocity dispersion in excess of 600 km/s. More compact line profiles would have been expected to be detected - albeit at low signal to noise ratio - in the shallower medium resolution observations. Furthermore, the source is remarkably compact, with photometry consistent with a point source. This places very interesting limits on the physical size for a galaxy at $z = 13$, with half-mass radius below 35 pc. The spectrum does not show any other spectral line that would unequivocally support the redshift measurement and the interpretation presented in the manuscript.

The results presented are unexpected, fascinating, and have potentially far-reaching consequences for understanding the earliest phases of cosmic reionization. How common (or rare) are sources like this with a topology of localised absorption of Ly α photons (that creates the red offset) and yet a globally significant escape fraction that enables the line to shine with high equivalent width? Is the combination of size and stellar mass of the source offering clues on what are the major sources for reionization? And is the Ly α emission powered by young very low metallicity stars, or by an AGN, possibly indicating we are observing a progenitor of the supermassive black holes seen as high- z QSOs?

While the manuscript has clear potential as one of the most significant JWST results so far for the study of the first generations of galaxies, I have various questions and comments on the data analysis and on the interpretation presented that would improve the work. These fall broadly into two major categories:

1) Robustness of the spectral line identification. The key question as a reader of the current manuscript is the statistical significance of the spectral line at observed wavelength ~ 1.7 micron.

(a) The line flux measurement is presented at SNR=6.4 from the final spectrum that combines two independent visits. Yet, the two independent visits are reported to have SNR=1.9 and 2.8 respectively (ref: Methods "Emission Line Properties" page 12). Thus, one would have expected the combined data to have SNR ~ 3.4 .

(b) The 2D spectrum in the top panel of Fig 1 visually shows the line standing out relative to the redwards continuum in about 3 resolution elements. The color scale shown, which is maxing out at SNR=2, would then indicate that the line has SNR $\sim \sqrt{3} \times 2 \sim 3.4$, consistent with the combination of SNR from the individual visits from point (a) above.

(c) The two independent visits have nearly identical microshutter placement, orientation and nodding strategy. I think this would mean that the line measurements originate from essentially the same small set of detector pixels in both visits. The manuscript would benefit from analysis of individual exposure frames to verify identification of any hot/warm pixel and

present evidence of robust data quality.

(d) In a recent paper by Boyett et al. (2024, NatAst) there is evidence of contamination of a $z=9.1$ galaxy spectrum by targets in separate shutters in the NIRSpec MSA with a similar row number. This manuscript would benefit from a similar analysis to exclude (or comment on) serendipitous association of the line with the continuum spectrum of the target.

(2) Interpretation of the line. Assuming the line is a physical feature of the spectrum and detected at sufficiently high confidence, the next key question is its interpretation.

(a) Given the photometry is unresolved and therefore consistent with a point source, the manuscript would benefit from a discussion of potential interpretation of the photometry and spectrum as a brown dwarf to demonstrate that such solution can be confidently ruled out. Incidentally, I note that a very quick literature search for IR spectroscopy of brown dwarfs brings up a paper that show a potential Al emission line at around 1.7 micron (Steele et al. 1995, MNRAS, 275, 841, cf Fig. 1).

(b) The identification of a single line as Ly α without supporting evidence from a second line detection is the point of the manuscript that has the greatest potential for improvement in its current form, given that the physical interpretation of such strong Ly α emission challenges the expectations of how reionization proceeds at $z>10$. The Methods section gives upper limits on the EW of other lines, with no detection reported (cf. page 13). What is the confidence level of these upper limits (e.g. 1, 2 or 3 sigma)? It would be very helpful to present the actual SNR measurements and/or zoom in on the spectrum at the expected line positions based on the best fit redshift instead of only quoting limits. I would also recommend to stack the spectrum at the expected locations of the other lines to identify any supporting evidence for the redshift inference and Ly α identification.

(c) Assuming sufficient confidence that the line is Ly α , the manuscript would benefit from greater discussion of the possibility that the spectrum is dominated by, or has substantial contribution from, an AGN. The potentially broad Ly α profile and the unresolved nature of the source would be hints supporting identification of the spectrum as an AGN, yet the manuscript only addresses this scenario in passing and without quantitative discussion. For example, based on other measurements of galaxy sizes for sources of similar luminosity at $z\sim 8$, how does the upper limit of <35 pc physical half-light radius compare to? And how would an AGN/QSO spectrum fit compare to the stellar population templates presented in the paper? In this respect, I note that the Method section highlights how the stellar population modeling assuming light from stellar populations returns a relatively poor quality of fit even for the best fit model, underscoring that there are systematic issues at play in the data-model comparison (or underestimation of the photometric errors).

(d) The stellar population modeling with BAGPIPES is carried out using photometric data. Would direct modelling of the spectrum give a consistent answer?

Prof. Michele Trenti
School of Physics
The University of Melbourne

Referee #2

(Remarks to the Author)

The paper presents the detection of Lyman alpha emission from a $z=13$ galaxy. The presence of the emission line is a new and interesting discovery as it suggests that young massive stars or AGN have created an ionized bubble around this system. This result is particularly significant for our understanding of galaxy formation and the implications of having processes that generate large amounts of Lyman continuum photons that can ionize the IGM.

The data analysis and inference are rigorous and comprehensive. I have suggestions regarding data presentation, which are addressed below. The statistical aspects covered in the paper seem robust.

The key conclusion is valid and reliable. However, a little more discussion on the mechanism and the implication of the size of the ionization bubble may be elaborated.

The paper can improve the presentation of the data as well in the discussion of the physical picture of this target that led to the observation of the Lyman alpha emission as the Lyman alpha damping wings. In particular, the plots are very dense, with insets and multiple panels. I suggest that the authors consider simplifying some of the plots (e.g. fig 2 and 6). Similarly, Fig 7 can be separated into two figures, i.e., a separate figure for the inset. The inference from the inset regarding the lookback time of 10 to -2 Myrs is unclear from the caption and labels.

Most importantly, the paper would benefit from a schematic showing the physical structure of the galaxy and possible orientations discussed in the text, such as foreground DLA at the same redshift and/or edge-on (inclined) galaxy, etc.

Referee #3

(Remarks to the Author)

Key results: The key result of this manuscript is the discovery and characterization of an interesting source in the early universe, $z \sim 13$, which exhibits the Lyman alpha line with a relatively high EW, indicating a prolific production of ionizing photons which are able to escape. The authors aim to constrain the physical properties and conditions of the source.

Originality and significance: The discovery is certainly original: this source is unique among the so-far-discovered sources at the same epoch. The significance of this discovery could benefit from further clarification, particularly given that it is a single source. A more detailed explanation of how the various pieces of data fit together may help strengthen the interpretation and provide a clearer understanding of the findings. Comparison is made with one specific object at $z \sim 6$ (that also exhibits strong emission) but not with other objects at the same redshift ($z > 10$) that provide a probe of the same epoch which can potentially shed light on why and how this particular source is able to ionize so much of its environment while others have failed.

Data & methodology: The core data appears to be proprietary, but the reduced spectra are presented (though see notes on plot readability below). One concern is that the pipeline used for this reduction is not public, nor is there a paper providing a detailed description of it. While a public pipeline with similar functionality exists, the team has opted to use a proprietary one. It would be helpful to understand why the public pipeline was not used and what advantages the proprietary version offers. For instance, custom clipping algorithms, mentioned as a difference, could also be implemented in the public pipeline. A comparison between the open-source pipeline and the one used in this paper would provide valuable context. That said, the data and methodology are described in great detail, with all steps of the analysis clearly outlined. In most sections, the authors use at least two different approaches to constrain the derived properties, with priors and assumptions clearly stated.

Appropriate use of statistics and treatment of uncertainties: Appropriate error bars are defined and included in all figures and tables. However a statement on page 14 (second paragraph) raises concerns about the analysis: "the comparatively high χ^2 (...) indicates that the photometric errors are potentially underestimated". Underestimation of the photometric errors would potentially impact the errors of the normalization of the spectrum to the photometry as well as the results from the SED fitting and derived parameters and their error bars. Considering the wealth of information in the larger dataset from which this source has been selected, can the authors quantify the level of underestimate of the photometric errors? How does increasing the errors on the photometry within some justified range impact the results of the SED fits and spectral modeling?

Conclusions: While the Methods section is well-executed, the conclusions and interpretation in the main body of the manuscript could be stronger. The authors present a detailed analysis of a unique and extraordinary object, but it's not immediately clear how well the pieces of the puzzle fit together, or what gaps remain if they don't. For example, the stellar population modeling and the spectral modeling separately constrain the same parameters (e.g., UV slope) but the fits are done independently, which despite the advantages, results in a lack of clear constraints. The discussion touches on several potential contributors to the observed spectrum (e.g., 2-photon continuum, supermassive black holes, central starburst, Pop III stars, high ISM density, and nebular emission), but lacks a clear conclusion that emphasizes the most likely interpretation of the physical conditions and radiation sources of the object. This uncertainty is also reflected in the abstract, which focuses on observational characteristics without addressing the physical interpretation.

Suggested improvements: The following changes can potentially strengthen the work:

Data reduction: the authors do not mention a bar shadow correction, is that implemented and what impact can that have on the spectrum (as this is one correction that can change the shape of the continuum)?

The 2-photon continuum in Figure 2b and in Figure 3d look different: are they both the same fits to the data? The continuum in figure 2b looks like a much worse fit to the data.

Stellar population synthesis modeling: the modeling performed is fairly standard while the source is anything but. The authors assume a pure stellar population (with and without nebular emission) with a standard IMF and fairly standard inputs and priors on all parameters. The final two models Table 2/Figure 6 fit equally well, despite having quite different properties. Are there changes in the inputs (specifically IMF) that would result in improvements to the fit? The discussion mentions 2-photon continuum, supermassive black holes and PopIII stars, as potential alternatives but these alternatives do not need to be fully responsible for all the flux. Have the authors considered composite models and is it possible to reconcile the contradicting constraints (with a lower χ^2) with such a model?

Improvements to figures: The figures are aesthetically beautiful. However, some of them are not readable in color and especially not in grayscale (which is an accessibility issue). Specific elements that are not readable: Figure 1 the ForcePro and CIRC2 datapoints are not clearly visible; Figure 2 the DLA and IGM transmissions are not clearly distinguishable; Figure 3, all panels, the different lines are not clearly distinguishable, and panel d is hard to read even in color; Figure 5a, hard to read even in color; Figure 5b - the points for the 2 different gratings are indistinguishable in grayscale. Figure 6a and b - the default and no nebular fits are indistinguishable in grayscale; Figure 7 inset - recommend switching the left and right axis.

References: Does this manuscript reference previous literature appropriately? If not, what references should be included or excluded? No recommendations on additional references.

Clarity and context: Is the abstract clear, accessible? Are abstract, introduction and conclusions appropriate? Yes, for the most part. Some parts of the text have overly long sentences and an abundance of parenthetical phrases which makes reading the text a bit cumbersome.

Version 2:

Reviewer comments:

Referee #1

(Remarks to the Author)

Referee report for manuscript 485218_2 - Witstock et al.

I thank the authors for addressing my comments and questions on the original manuscript and for their answers and edits in the text of the revised version. I consider the revised text clear, original and engaging to read. I am also fully satisfied with the responses received to my data reduction queries and I have no further substantive comments to improve this very interesting manuscript.

Two minor comments:

I suggest checking the flow of the sentence at the end of the second to last paragraph of the main text on page 4 ("Constraints... a viable candidate.")

I suggest clarifying the caption of Extended Data Fig. 3 on page 9 to specify that the annotation of the Key UV emission lines represent expected locations, with the spectrum showing no evidence of any detection.

Prof. Michele Trenti
School of Physics
The University of Melbourne

Referee #2

(Remarks to the Author)

I am happy with the new additions. I recommend the manuscript for publication.

Referee #3

(Remarks to the Author)

The resubmitted manuscript adequately addresses all issues raised in the initial review. The new focus on the understanding the physical properties of the source is very good. I appreciate the inclusion of the new Figure 3. I would however recommend that the authors split the 2 possible scenarios in 2 separate panels in this figure. The current figure seems to represent two different sources, an AGN and a massive stellar pop embedded in a single ionized bubble and superimposed along the line of sight to the observer. This does not seem to be the intent, they are supposed to be 2 different alternatives.

Other than that, I recommend the manuscript for publication.

Response to referee reports

Nature manuscript 2024-07-13969A

General remarks

Firstly, we would like to thank the referees for their careful reading of the manuscript and their suggestions, which have been helpful in improving the manuscript. We have made a number of clarifications in the text and considered additional models, added a new figure presenting a schematic of the physical structure of the galaxy and another figure in the Methods with a thorough statistical analysis of the emission line. Changes to the manuscript are shown in the diff.pdf file. Below, we respond to individual remarks by the referees in black.

Relating to points raised about the NIRSpec data reduction, we wish to emphasise that the official, publicly available STScI pipeline software consists of the same core algorithms that are present in the GTO pipeline used in the current work (outlined in Ferruit et al. 2022, A&A, 661, A81). What has now evolved into the GTO pipeline was originally delivered to NASA together with the NIRSpec instrument by the European Space Agency (ESA), which however does not wish to release their version of the pipeline for reasons pertaining to software licensing, user support, documentation, and maintenance. As suggested by referee #3, we have directly tested the consistency between the two pipelines by reducing our data with the default Space Telescope Science Institute (STScI) pipeline. The latter provides a consistent, though generally noisier spectrum (even if the estimated uncertainty is less conservative). We note in particular that the emission line at $1.7\ \mu\text{m}$ is detected with a fully consistent flux level (at an estimated SNR = 8.5) according to the spectrum produced by the STScI pipeline (compared to SNR = 6.5 for the GTO pipeline).

The key difference with the default STScI pipeline lies in the combination process of all individual exposures and error estimation and propagation, which we now discuss in greater detail in the Methods. Specifically, we use a refined sigma-clipping procedure, first introduced in Hainline et al. (2024b, ApJ, 976, 160), that differs minimally from the standard combination procedure adopted by the GTO pipeline (outlined in e.g. Bunker et al. 2024, A&A, 690, A288). The standard pipeline was designed to give a conservative uncertainty estimate in all cases, including when only a limited number of exposures are available, and provides an 'effective' error spectrum that is meant to represent statistically independent uncertainties on individual wavelength bins. This is achieved by scaling up the error spectrum to capture the (significant) covariance between adjacent wavelength bins. The refined approach presented in Hainline et al. (2024b) incorporates a bootstrapping procedure that allows us to also obtain a full covariance matrix via bootstrapping, as also discussed in Curti et al. (2024, arXiv240702575), which directly captures all sources of noise present in the data.

Referee #1 (Remarks to the Author):

The manuscript presents deep low-resolution JWST NIRSpec observations of a photometric candidate galaxy at redshift $z \sim 13$, with the analysis showing a clear continuum break in the spectrum and - remarkably - an emission line bluewards of the spectral break, which is identified as Ly α . The results presented are unique since none of the handful of galaxies spectroscopically confirmed at $z > 10$ have shown Ly α emission, as naturally expected because of the largely ionized nature of the intergalactic medium at those early times. The Ly α emission peak detected in this source appears to fall substantially redwards of the systemic galaxy redshift (velocity offset greater than 500 km/s). Also, the observations are at low spectral resolution yet there is some evidence presented to support a broadened line profile, with velocity dispersion in excess of 600 km/s. More compact line profiles would have been expected to be detected - albeit at low signal to noise ratio - in the shallower medium resolution observations. Furthermore, the source is remarkably compact, with photometry consistent with a point source. This places very interesting limits on the physical size for a galaxy at $z = 13$, with half-mass radius below 35 pc. The spectrum does not show any other spectral line that would unequivocally support the redshift measurement and the interpretation presented in the manuscript.

The results presented are unexpected, fascinating, and have potentially far-reaching consequences for understanding the earliest phases of cosmic reionization. How common (or rare) are sources like this with a topology of localised absorption of Ly α photons (that creates the red offset) and yet a globally significant escape fraction that enables the line to shine with high equivalent width? Is the combination of size and stellar mass of the source offering clues on what are the major sources for reionization? And is the Ly α emission powered by young very low metallicity stars, or by an AGN, possibly indicating we are observing a progenitor of the supermassive black holes seen as high- z QSOs?

While the manuscript has clear potential as one of the most significant JWST results so far for the study of the first generations of galaxies, I have various questions and comments on the data analysis and on the interpretation presented that would improve the work. These fall broadly into two major categories:

1) Robustness of the spectral line identification. The key question as a reader of the current manuscript is the statistical significance of the spectral line at observed wavelength ~ 1.7 micron.

(a) The line flux measurement is presented at SNR=6.4 from the final spectrum that combines two independent visits. Yet, the two independent visits are reported to have SNR=1.9 and 2.8 respectively (ref: Methods "Emission Line Properties" page 12). Thus, one would have expected the combined data to have SNR ~ 3.4 .

We thank the referee for pointing out this apparent inconsistency in the estimated total SNR compared to the two independent visits. We first wish to clarify that the spectra obtained

from individual visits were reduced via the standard pipeline 5-pixel extraction, as opposed to the 3-pixel extraction we otherwise preferred for this very compact source (Spectroscopic measurements) with refined sigma-clipping procedure (see above), which explains the discrepancy in the SNR quoted from the fiducial ('sigma-clipped') spectrum (SNR = 6.4).

To address this issue, apart from the improved discussion of the combination process and uncertainty estimation described above, we now quote figures from the two separate visits consistently extracted from 3 pixels and processed following the refined procedure, which result in SNR = 4.2 (visit 1) and SNR = 5.0 (visit 3), perfectly in agreement with the overall SNR = 6.4.

Additionally, we have implemented a more thorough analysis of the 1D spectra from all individual exposures, presented in a new figure. This clearly demonstrates that (1) our fiducial sigma-clipped spectrum and covariance matrix obtained via the bootstrapping procedure provide excellent agreement with the mean flux and associated uncertainty estimated through a jackknife resampling method and (2) we detect the line at SNR > 3 in each of four contiguous wavelength bins individually, including at SNR > 5 in the central two. The SNR of the integrated line flux across these four bins (SNR = 6.4, both for the jackknife and the bootstrapped covariance matrix estimates) is not substantially increased, as a result of the large covariance present between adjacent wavelength bins.

(b) The 2D spectrum in the top panel of Fig 1 visually shows the line standing out relative to the redwards continuum in about 3 resolution elements. The color scale shown, which is maxing out at SNR=2, would then indicate that the line has $\text{SNR} \sim \sqrt{3} \times 2 \sim 3.4$, consistent with the combination of SNR from the individual visits from point (a) above.

In this case, it should have been noted that the 1D spectra are always extracted from individual exposures and combined afterwards, while the combination of 2D spectra is done separately (see details in Bunker et al. 2024, A&A, 690, A288). Importantly, this means the final combined 1D spectrum is not a direct extraction from the 2D map. The 2D spectrum, less accurate and rigorous in terms of combination of the spectra, is shown mainly for visualisation purposes, while the 1D spectrum provides the more rigorous and accurate quantitative information. This has now been clarified in the text.

(c) The two independent visits have nearly identical microshutter placement, orientation and nodding strategy. I think this would mean that the line measurements originate from essentially the same small set of detector pixels in both visits. The manuscript would benefit from analysis of individual exposure frames to verify identification of any hot/warm pixel and present evidence of robust data quality.

By design, the three-point nodding pattern ensures that the sub-exposures, and therefore the line measurements, do not originate from the same set of detector pixels. Moreover, the microshutters employed are slightly different between the two visits (3, 237, 87 and 3, 237, 86). Together, this leads to a vertical shift of ~15 pixels on the detector between individual exposures. Finally, from a statistical analysis of the individual exposures there is no sign of any significant impact from noisy pixels and/or cosmic ray artefacts. This has now been discussed in the text.

(d) In a recent paper by Boyett et al. (2024, NatAst) there is evidence of contamination of a $z=9.1$ galaxy spectrum by targets in separate shutters in the NIRSpec MSA with a similar row number. This manuscript would benefit from a similar analysis to exclude (or comment on) serendipitous association of the line with the continuum spectrum of the target.

We confirm that we have visually inspected the exposures for any potential contamination of the spectra (both from separate open shutters or potential overlapping targets), which has now been described in the text. For completeness, we include cutouts of annotated raw exposures (before rectification, background subtraction, etc.) for the two visits here, where it can be seen the spectral traces of JADES-GS-z13-1-LA (NIRSpec target ID 20013731) are free of contamination:

(2) Interpretation of the line. Assuming the line is a physical feature of the spectrum and detected at sufficiently high confidence, the next key question is its interpretation.

(a) Given the photometry is unresolved and therefore consistent with a point source, the manuscript would benefit from a discussion of potential interpretation of the photometry and spectrum as a brown dwarf to demonstrate that such solution can be confidently ruled out. Incidentally, I note that a very quick literature search for IR spectroscopy of brown dwarfs brings up a paper that show a potential Al emission line at around 1.7 micron (Steele et al. 1995, MNRAS, 275, 841, cf Fig. 1).

We agree that we should discuss the brown dwarf scenario, which has now been implemented in the text. Indeed, this has already been confidently ruled out in works by Hainline et al. (2024a, ApJ, 964, 71) and (based on the more extensive photometric data set also used in the current work) by Robertson et al. (2024, ApJ, 970, 31). In short, the observed photometry of JADES-GS-z13-1-LA consists of stringent upper limits below ~ 1.7 micron combined with a blue continuum redwards, which is inconsistent with the SEDs of

brown dwarfs that show a number of strong molecular atmospheric absorption features at 1 - 5 microns. We include a brown dwarf model fit (using ATMO2020++, Philips et al. 2020, A&A, 637, A38; Meisner et al. 2023, AJ, 166, 57) to the photometry of JADES-GS-z13-1-LA below, showing very significant discrepancies across the SED:

(b) The identification of a single line as Ly α without supporting evidence from a second line detection is the point of the manuscript that has the greatest potential for improvement in its current form, given that the physical interpretation of such strong Ly α emission challenges the expectations of how reionization proceeds at $z > 10$. The Methods section gives upper limits on the EW of other lines, with no detection reported (cf. page 13). What is the confidence level of these upper limits (e.g. 1, 2 or 3 sigma)? It would be very helpful to present the actual SNR measurements and/or zoom in on the spectrum at the expected line positions based on the best fit redshift instead of only quoting limits. I would also recommend to stack the spectrum at the expected locations of the other lines to identify any supporting evidence for the redshift inference and Ly α identification.

We agree with the referee that ideally, we would have access to multiple spectral features to confirm the systemic redshift of this object. Nevertheless, we wish to emphasise that there already is an independent spectral feature, the strong Lyman- α break, that independently secures confirmation of the systemic redshift and therefore of the identification of the line as Ly α . When damping-wing absorption is properly taken into account, the break is a reliable means of measuring the systemic redshift (and indeed the only significant spectral feature supporting many confirmed $z > 10$ galaxies so far), as recently demonstrated by the ALMA detection of [OIII] 88 μ m in JADES-GS-z14-0 (Schouws et al. 2024, arXiv:2409.20533).

The upper limits on all emission lines quoted in the manuscript are 3σ (now summarised in a table in the Methods). We have repeated the analysis detailed in the appendices of Hainline et al. (2024b, ApJ, 976, 160) and Carniani et al. (2024, Nature, 633, 318), in which the inferred the one-sided p-value for a set of different emission lines is combined to yield the statistical significance of a given spectroscopic redshift over a range of $\Delta z = 0.2$ centred on z

= 13.0. Essentially, this corresponds to stacking the spectrum at the expected location of different emission lines for a given redshift, yet does not penalise the potential absence of any of them. The results of this analysis (included below) do not show any significant peaks, in agreement with our reported findings that no other emission lines are significantly detected. This has now been discussed in the text.

(c) Assuming sufficient confidence that the line is Ly α , the manuscript would benefit from greater discussion of the possibility that the spectrum is dominated by, or has substantial contribution from, an AGN. The potentially broad Ly α profile and the unresolved nature of the source would be hints supporting identification of the spectrum as an AGN, yet the manuscript only addresses this scenario in passing and without quantitative discussion. For example, based on other measurements of galaxy sizes for sources of similar luminosity at $z > \sim 8$, how does the upper limit of < 35 pc physical half-light radius compare to? And how would an AGN/QSO spectrum fit compare to the stellar population templates presented in the paper? In this respect, I note that the Method section highlights how the stellar population modeling assuming light from stellar populations returns a relatively poor quality of fit even for the best fit model, underscoring that there are systematic issues at play in the data-model comparison (or underestimation of the photometric errors).

We agree with the referee that the possibility of an AGN warrants more detailed discussion. This has now been implemented as part of the restructuring of the main text. Based on a comparison of its size, broad Ly α line, blue UV slope, and likely presence of an ionised bubble, we conclude that an AGN may be a viable scenario, although there is no conclusive evidence in favour or against it. As argued in the text, we opted for performing our main spectral fitting with a more flexible power-law model, and refrained from doing so directly with QSO templates, since these are primarily metal-rich, reddened objects displaying strong UV emission lines (e.g. the composite spectrum of Vanden Berk et al. 2001, AJ, 122, 549 has a CIV EW of 27 Å and CIII EW of 21 Å), in tension with the observed blue spectrum and absence of other emission lines.

(d) The stellar population modeling with BAGPIPES is carried out using photometric data. Would direct modelling of the spectrum give a consistent answer?

The stellar population synthesis models are fitted to photometry and the spectrum simultaneously, where the photometry only provides a normalisation. They are in good agreement with a newly presented BEAGLE fit to the spectrum only, where the escape fraction is moreover freely varied. We also note our results (M_* and SFR) are consistent within 1σ compared to results with a third SED fitting code, Prospector, presented in Robertson et al. (2024, ApJ, 970, 31) that were fit only to photometry.

Referee #2 (Remarks to the Author):

The paper presents the detection of Lyman alpha emission from a $z=13$ galaxy. The presence of the emission line is a new and interesting discovery as it suggests that young massive stars or AGN have created an ionized bubble around this system. This result is particularly significant for our understanding of galaxy formation and the implications of having processes that generate large amounts of Lyman continuum photons that can ionize the IGM.

The data analysis and inference are rigorous and comprehensive. I have suggestions regarding data presentation, which are addressed below. The statistical aspects covered in the paper seem robust.

The key conclusion is valid and reliable. However, a little more discussion on the mechanism and the implication of the size of the ionization bubble may be elaborated.

We now briefly discuss the implications on reionisation as a whole in the main text and we have clarified the discussion on the creation of the ionised bubble in the Methods.

The paper can improve the presentation of the data as well in the discussion of the physical picture of this target that led to the observation of the Lyman alpha emission as the Lyman alpha damping wings. In particular, the plots are very dense, with insets and multiple panels. I suggest that the authors consider simplifying some of the plots (e.g. fig 2 and 6). Similarly, Fig 7 can be separated into two figures, i.e., a separate figure for the inset. The inference from the inset regarding the lookback time of 10 to -2 Myrs is unclear from the caption and labels.

We agree with the referee that the clarity of some of the figures suffered in our attempt to convey all relevant information in the limited space (further addressed in the response to referee #3 below). We have implemented the suggested simplification and other changes to these figures.

Most importantly, the paper would benefit from a schematic showing the physical structure of the galaxy and possible orientations discussed in the text, such as foreground DLA at the same redshift and/or edge-on (inclined) galaxy, etc.

We thank the referee for this suggestion, which we agree benefits the discussion in the main text. A schematic has now been included and discussed in the manuscript.

Referee #3 (Remarks to the Author):

Key results: The key result of this manuscript is the discovery and characterization of an interesting source in the early universe, $z \sim 13$, which exhibits the Lyman alpha line with a relatively high EW, indicating a prolific production of ionizing photons which are able to escape. The authors aim to constrain the physical properties and conditions of the source.

Originality and significance: The discovery is certainly original: this source is unique among the so-far-discovered sources at the same epoch. The significance of this discovery could benefit from further clarification, particularly given that it is a single source. A more detailed explanation of how the various pieces of data fit together may help strengthen the interpretation and provide a clearer understanding of the findings. Comparison is made with one specific object at $z \sim 6$ (that also exhibits strong emission) but not with other objects at the same redshift ($z > 10$) that provide a probe of the same epoch which can potentially shed light on why and how this particular source is able to ionize so much of its environment while others have failed.

As further detailed below, we have implemented an extensive restructuring of the main text, as well as the inclusion of a schematic (see above), to better clarify what is the current observational evidence and how this can be interpreted.

Data & methodology: The core data appears to be proprietary, but the reduced spectra are presented (though see notes on plot readability below). One concern is that the pipeline used for this reduction is not public, nor is there a paper providing a detailed description of it. While a public pipeline with similar functionality exists, the team has opted to use a proprietary one. It would be helpful to understand why the public pipeline was not used and what advantages the proprietary version offers. For instance, custom clipping algorithms, mentioned as a difference, could also be implemented in the public pipeline. A comparison between the open-source pipeline and the one used in this paper would provide valuable context. That said, the data and methodology are described in great detail, with all steps of the analysis clearly outlined. In most sections, the authors use at least two different approaches to constrain the derived properties, with priors and assumptions clearly stated.

As discussed in the General Remarks, the revised manuscript now makes a direct comparison with the public STScI pipeline reduction, and includes a more detailed description of the main sigma-clipping algorithm adopted here (introduced in Hainline et al. 2024b, ApJ, 976, 160). Moreover, we are more than happy to provide reduced spectra (including of all individual sub-exposures) upon publication.

Appropriate use of statistics and treatment of uncertainties: Appropriate error bars are defined and included in all figures and tables. However a statement on page 14 (second paragraph) raises concerns about the analysis: “the comparatively high χ^2 (...) indicates that the photometric errors are potentially underestimated”. Underestimation of the photometric errors would potentially impact the errors of the normalization of the spectrum to

the photometry as well as the results from the SED fitting and derived parameters and their error bars. Considering the wealth of information in the larger dataset from which this source has been selected, can the authors quantify the level of underestimate of the photometric errors? How does increasing the errors on the photometry within some justified range impact the results of the SED fits and spectral modeling?

We thank the referee for pointing out this issue. As discussed in the Photometric measurements section, the ForcePho fitting is performed on individual NIRCcam exposures, and therefore it entirely bypasses the correlated noise introduced by the mosaicing process. Furthermore, substantial effort has gone into optimising the background subtraction and quantifying the noise properties of the NIRCcam imaging (e.g. Rieke et al. 2023, ApJS, 269, 16). Nevertheless, the extracted photometric fluxes may still suffer from a degree of systematic uncertainty, including from any imperfections in the flat fields, an effect that is difficult to quantify directly for a single source. This is taken into account empirically by the aperture ('CIRC2') photometry presented separately, where the estimated uncertainty considers the scatter found in a number of randomly placed empty apertures. However, the aperture photometry is performed on the mosaiced images and is therefore impacted by correlated noise. We find that the combination of nominal ForcePho and CIRC2 uncertainties alleviates the issue of high χ^2 values. Furthermore, the comparison between NIRSpect and NIRCcam (spectro)photometry does not show any wavelength dependency and is entirely consistent between the ForcePho and aperture photometry, implying additional systematic uncertainty does not impact the additional path-loss corrections. Similarly, we now show that the complete exclusion of the (normalisation to) photometric data does not significantly impact the findings in the SED fitting, as discussed above. These considerations have now been implemented in the text.

Conclusions: While the Methods section is well-executed, the conclusions and interpretation in the main body of the manuscript could be stronger. The authors present a detailed analysis of a unique and extraordinary object, but it's not immediately clear how well the pieces of the puzzle fit together, or what gaps remain if they don't. For example, the stellar population modeling and the spectral modeling separately constrain the same parameters (e.g., UV slope) but the fits are done independently, which despite the advantages, results in a lack of clear constraints. The discussion touches on several potential contributors to the observed spectrum (e.g., 2-photon continuum, supermassive black holes, central starburst, Pop III stars, high ISM density, and nebular emission), but lacks a clear conclusion that emphasizes the most likely interpretation of the physical conditions and radiation sources of the object. This uncertainty is also reflected in the abstract, which focuses on observational characteristics without addressing the physical interpretation.

We agree with the referee that this point deserves careful reconsideration. As discussed above, we have implemented an extensive restructuring of the main text, as well as the inclusion of a schematic (see above), to better clarify what is the current observational evidence and how this can be interpreted. Our adopted approach is to focus on our custom spectral modelling which is able to simultaneously model the emission line and continuum, where we adopt a power law to maintain flexibility. This is mainly due to the inability of commonly used SED fitting codes to self-consistently model Ly α emission, let alone its coexistence with a smooth spectral turnover as observed. Given the signal-to-noise ratio offered by the current data, we moreover cannot confidently detect, and hence model,

spectral features other than the Ly α line and UV continuum, including the strong turnover. While the exact nature of the object therefore cannot yet be definitively established and will require follow-up observations – as is the case for other galaxies with similar features, such as GS-9422 that has been extensively debated in the literature – we can constrain specific physical parameters and rule out certain scenarios. A newly included BEAGLE model shows the data strongly favour a high LyC escape fraction, which is independently confirmed in a new spectral model where we consider the more general combination of a power law continuum and nebular spectrum tied to the intrinsic strength of Ly α (see below). These considerations and motivations are now clarified and/or newly implemented in the text.

Suggested improvements: The following changes can potentially strengthen the work:
Data reduction: the authors do not mention a bar shadow correction, is that implemented and what impact can that have on the spectrum (as this is one correction that can change the shape of the continuum)?

Our initial path-loss corrections are calculated by modelling the full optical path through the NIRSpec instrument for a point source located at the intra-shutter location of the source, thereby taking into account all relevant instrumental effects including the vignetting due to the bar shadow. We also performed additional path-loss corrections based on the NIRCам photometry, where we do not find a wavelength dependence. While we do not explicitly perform additional path-loss corrections for *extended* sources due to the bar shadow, given its very compact morphology GS-z13-1-LA effectively is a point source at the spatial resolution of NIRCам and NIRSpec, and hence this effect should be negligible.

The 2-photon continuum in Figure 2b and in Figure 3d look different: are they both the same fits to the data? The continuum in figure 2b looks like a much worse fit to the data.

The 2-photon continuum shown in Figure 3d is not a fit to the data (nor is it convolved with the LSF), but like the spectrum of GS-9422 it is simply normalised to the spectrum of GS-z13-1-LA at a rest-frame wavelength of 1500 Å. The lower normalisation seen in Figure 2b, although not fitting quite as well around the turnover, provides smaller residuals at longer wavelengths, which is due to the fixed shape of the 2-photon continuum not being quite as blue as the observed spectrum. Overall, it does therefore result in a slightly better fit to the spectrum.

Stellar population synthesis modeling: the modeling performed is fairly standard while the source is anything but. The authors assume a pure stellar population (with and without nebular emission) with a standard IMF and fairly standard inputs and priors on all parameters. The final two models Table 2/Figure 6 fit equally well, despite having quite different properties. Are there changes in the inputs (specifically IMF) that would result in improvements to the fit? The discussion mentions 2-photon continuum, supermassive black holes and PopIII stars, as potential alternatives but these alternatives do not need to be fully responsible for all the flux. Have the authors considered composite models and is it possible to reconcile the contradicting constraints (with a lower χ^2) with such a model?

Further to the discussion above, the manuscript now contains more discussion on the shortcomings of standard stellar models. In particular, the necessity of a very high escape fraction in such models implies that the observed continuum (to good approximation) follows

a simple power law. This extends to top-heavy IMFs and PopIII stars, since these actually are predicted to have (much) redder UV slopes in the presence of nebular continuum. The power-law scenario also holds for an AGN continuum and is incorporated in our fiducial model that so far provides the best fit to the current data. Having said that, we have verified a composite model with a freely varying power law and two-photon continuum components provides a similar, but slightly higher χ^2 (not currently included in the current manuscript). Finally, a newly included spectral model where we consider the combination of a power law continuum and nebular spectrum that is self-consistently tied to the Ly α luminosity again prefers a high LyC escape fraction, though does not improve the χ^2 .

Improvements to figures: The figures are aesthetically beautiful. However, some of them are not readable in color and especially not in grayscale (which is an accessibility issue). Specific elements that are not readable: Figure 1 the ForcePro and CIRC2 datapoints are not clearly visible; Figure 2 the DLA and IGM transmissions are not clearly distinguishable; Figure 3, all panels, the different lines are not clearly distinguishable, and panel d is hard to read even in color; Figure 5a, hard to read even in color; Figure 5b - the points for the 2 different gratings are indistinguishable in grayscale. Figure 6a and b - the default and no nebular fits are indistinguishable in grayscale; Figure 7 inset - recommend switching the left and right axis.

We thank the referee for pointing out these issues. We have implemented the suggested changes for all figures throughout the manuscript and ensured they are readable without colour.

References: Does this manuscript reference previous literature appropriately? If not, what references should be included or excluded? No recommendations on additional references.

Clarity and context: Is the abstract clear, accessible? Are abstract, introduction and conclusions appropriate? Yes, for the most part. Some parts of the text have overly long sentences and an abundance of parenthetical phrases which makes reading the text a bit cumbersome.

As part of the restructuring and other revisions we have aimed to be as concise as possible, but we welcome recommendations for further improvement.

Response to editorial and referee comments

Nature manuscript 2024-07-13969A

Dear Dr Chadayammuri,

I am grateful for the positive and favourable response of the referees and yourself with regards to our resubmitted manuscript. To accommodate the editorial requirements, as per your suggestion we have moved the majority of the original Methods section to a new, separate section of Supplementary Information. Specifically, what previously were Extended Data Fig. 1, 3, 4, 5, and 7, and Extended Data Table 3 (together with much of the text describing these items) have now been removed from the Methods section. Other, very minimal changes to address the final comments by the referees are shown in the diff.pdf file (below, we respond to individual remarks in black).

I hope that I have included all necessary attachments and forms. Please let us know if any further edits are required. Looking forward to hearing from you.

Kind regards,
Joris Witstok

Referee #1 (Remarks to the Author)

Referee report for manuscript 485218_2 - Witstok et al.

I thank the authors for addressing my comments and questions on the original manuscript and for their answers and edits in the text of the revised version. I consider the revised text clear, original and engaging to read. I am also fully satisfied with the responses received to my data reduction queries and I have no further substantive comments to improve this very interesting manuscript.

Two minor comments:

I suggest checking the flow of the sentence at the end of the second to last paragraph of the main text on page 4 ("Constraints... a viable candidate.")

We thank the referee for picking up on this – the flow of the sentence has now been improved through a slight rephrasing.

I suggest clarifying the caption of Extended Data Fig. 3 on page 9 to specify that the annotation of the Key UV emission lines represent expected locations, with the spectrum showing no evidence of any detection.

The caption has been clarified.

Prof. Michele Trenti
School of Physics
The University of Melbourne

Referee #2 (Remarks to the Author):

I am happy with the new additions. I recommend the manuscript for publication.

Referee #3 (Remarks to the Author):

The resubmitted manuscript adequately addresses all issues raised in the initial review. The new focus on the understanding the physical properties of the source is very good. I appreciate the inclusion of the new Figure 3. I would however recommend that the authors split the 2 possible scenarios in 2 separate panels in this figure. The current figure seems to represent two different sources, an AGN and a massive stellar pop embedded in a single ionized bubble and superimposed along the line of sight to the observer. This does not seem to be the intent, they are supposed to be 2 different alternatives.

Other than that, I recommend the manuscript for publication.

We thank the referee for this suggestion, which we have implemented.